# Hell or High Water: Evaluating Agentic Recovery from External Failures

**Andrew Wang**   **Sophia Hager**   **Adi Asija**   **Daniel Khashabi**[*]   **Nicholas Andrews**[*]
Johns Hopkins University
{awang116, danielk, noa}@jhu.edu

## Abstract

As language model agents are applied to real world problems of increasing complexity, they will be expected to formulate plans across large search spaces. If those plans fail for reasons *beyond their control*, how well do language agents search for alternative ways to achieve their goals? We devise a specialized agentic planning benchmark to study this question. Each planning problem is solved via combinations of function calls. The agent searches for relevant functions from a set of over four thousand possibilities, and observes *environmental feedback* in the form of function outputs or error messages. Our benchmark confronts the agent with *external failures* in its workflow, such as functions that suddenly become unavailable. At the same time, even with the introduction of these failures, we guarantee that the task remains solvable. Ideally, an agent's performance on the planning task should not be affected by the presence of external failures. Overall, we find that language agents struggle to formulate and execute backup plans in response to environment feedback. While state-of-the-art models are often able to identify the correct function to use in the right context, they struggle to adapt to feedback from the environment and often fail to pursue alternate courses of action, even when the search space is artificially restricted. We provide a systematic analysis of the failures of both open-source and commercial models, examining the effects of search space size, as well as the benefits of scaling model size in our setting. Our analysis identifies key challenges for current generative models as well as promising directions for future work.[1]

## 1 Introduction

Large language models (LLMs) are capable of reasoning and planning through natural language. As a result, they now form the backbone of autonomous *language model agents* that follow user instructions and carry out actions in the real world. Increasingly, these LLM-driven agents are applied to complex or open-ended tasks, such as navigating the internet or automating tasks across a user's computer desktop environment.

Naturally, faults may arise that prevent a plan from working; we categorize faults into two broad classes: (1) *internal errors*, which arise from mistakes an agent makes such as passing the wrong arguments to a function, and (2) *external errors* that occur for reasons beyond an agent's control. As an example of the latter, a flight booking API may not have flights fulfilling the desired criteria, requiring the LLM agent to try alternate carrier booking APIs. In spite of these obstacles, users generally expect an agent to complete tasks come hell or high water.[2] Prior works have focused on studying the ability of language agents to correct their own mistakes (Shinn et al., 2023; Madaan et al., 2023; Chen et al., 2023a). In contrast,

---

[*]Equal advising.
[1]Code and data available here: https://github.com/JHU-CLSP/hell-or-high-water
[2]The phrase "hell or high water" denotes an obligation that should be fulfilled regardless of circumstances beyond one's control.

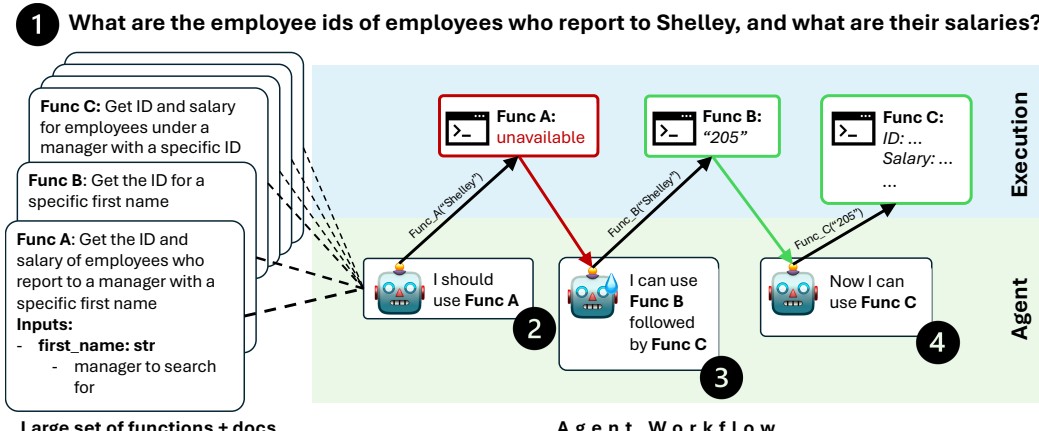

Figure 1: The agent correctly tries to use Func A, but encounters an *external error* outside its control: Func A is unavailable. Therefore, it must form a backup plan to use the composition of Func B and Func C. We include the full trajectory in Appendix F. **Language model agents struggle to identify backup plans when they encounter external errors.**

the effect of external faults has remained understudied, despite their prevalence in the real world.

We propose the first benchmark that studies the effect of external errors on language agent performance. Our benchmark satisfies a number of properties that make it suitable for studying this phenomenon. First, our evaluation consistently places an agent in a situation where the task cannot be completed without using a backup plan, while ensuring that the task remains solvable. Secondly, tasks in our benchmark are challenging enough to require planning, but straightforward enough that a successful backup plan can be found by interacting with the environment. Furthermore, to actually measure the inherent capabilities of language agents, our tasks are novel relative to the LLM pre-training data, avoiding solutions based on memorized behaviors. Finally, solutions are easily evaluated based on functional correctness—an agent's answer must match a gold solution. To satisfy this last requirement, we avoid relying on LLM verification to avoid potential biases and to facilitate reproducibility, and instead employ human-curated datasets where functional correctness can be automatically checked.

Figure 1 shows an example problem from the benchmark developed as part of this work. Given a question and access to a library of functions for querying and manipulating the relevant knowledge bases, the language agent must formulate a plan—a sequence of one or more function calls—and execute it to obtain the correct answer. Doing so involves not only formulating queries to retrieve the appropriate functions, but also calling these in the right order and with the right arguments. When the language agent executes a function, it is interacting with an external environment, which can respond in unexpected ways. Upon receiving unexpected feedback from the environment (e.g., if the function was called with the wrong arguments, or was unexpectedly disabled), a successful language agent must react accordingly by adjusting its plan. Our benchmark is constructed such that each question can be solved in at least two different ways, involving different functions.

In more detail, our proposed benchmark, *Hell-or-High-Water*, consists of 830 questions and 4450 functions that must be executed in appropriate contexts to obtain the right answer. Our benchmark creation process is completely automatic, converting text-to-SQL datasets into questions and tools. To consistently force a model to resort to a backup plan, we construct these functions such that every question in the dataset can be solved using either of two non-overlapping sequences of function calls. The first function an agent calls to attempt at solving the problem can be automatically disabled, while always leaving possible alternatives to arrive at the correct solution. While problems in our benchmark are straightforward to solve, involving no more than 3 reasoning steps, the presence of so many functions requires the

agent to plan. As agents are calling functions to generate the output, rather than generating a natural language output, functional correctness is straightforward for this benchmark.

Using the proposed benchmark, we conduct an extensive empirical study of language model agents which focuses on the following main questions:

- **Are language agents capable of fault-tolerant planning?** All LLMs evaluated have significant difficulty recovering from unexpected errors, including commercial models such as GPT-4 and Gemini 2.0.
- **How does size of the search space impact fault-tolerant planning?** We compare performance in two distinct settings: (1) when all required functions are included in-context; (2) when the language model agent must generate search queries to retrieve relevant tools. We find that the need to search for relevant tools results in significantly harder challenge.
- **Is fault-tolerant planning a property of model size?** While larger models perform better in terms of absolute accuracy, all models tested displayed decreased performance in the face of external errors, suggesting that resilience to such errors benefits weakly from model scaling.
- **Can language agents recognize when no solution is available?** In cases where all solutions are disabled (i.e., are present but return a clear error message), we find that different LLMs exhibit markedly different ability to recognize the lack of feasible solution, with some exhausting their turn budget.
- **What is the impact of question underspecification?** In the real-world, users may fail to adequately express their instructions, leading to underspecification. We find that the degree of question underspecification does impact results, with disambiguating information significantly helping certain models.

## 2 Related Work

**Real-world planning benchmarks.** A variety of benchmarks have previously been proposed to test LLM capabilities for real-world planning, with varying levels of complexity between benchmarks. Some, such as API-bank (Li et al., 2023) and ToolQA (Zhuang et al., 2023) specifically test a model's ability to plan, retrieve, and call a tool in a single step. Others, such as ToolSandbox (Lu et al., 2024a) and AppWorld (Trivedi et al., 2024) extend those capabilities by adding feedback, allowing for multi-turn interactive dialogues. TravelPlanner (Xie et al., 2024) requires agents to arrange a multi-day travel itinerary, which necessarily requires LLM agents to be able to navigate many multi-step tasks and compose outputs. In our benchmark, we provide multiple distinct solutions for each question with *different levels of complexity*: one solution solves the question in a single step, while the other requires the LLM to break the task into subtasks and call the correct function for each. Having a benchmark with multiple solutions, one of which requires more creativity and reasoning from the model, allows us to investigate an LLM's ability to adapt according to feedback. To our knowledge, prior benchmarks do not provide these types of problems.

**Tool use.** LLMs with the capability to use tools gain significant advantages, such as access to specialized knowledge and multimodal functionalities (Qin et al., 2023b). In particular, many systems have been designed to use tools accessed through APIs. Using APIs remains challenging for LLMs to use, as LLMs may fail to correctly call an API on the first attempt (Patil et al., 2023; Qin et al., 2023b). While many methods for creating an agent which can use tools require fine-tuning an LLM (Parisi et al., 2022; Patil et al., 2023; Qin et al., 2023b; Schick et al., 2023; Yang et al., 2023), modern LLMs have strong in-context learning abilities that allow for few-shot tool-use (Hao et al., 2023; Qin et al., 2023a; Lu et al., 2024b). In particular, frameworks such as ReACT (Yao et al., 2023) and CodeAct (Wang et al., 2024b) use alternating reasoning and action steps to allow LLM agents to receive feedback and adjust their use of tools accordingly; for instance, receiving an error message for a failed API call during a reasoning step may result in the model adjusting the call and correctly using the tool. Multi-step reasoning processes also allows for tool *composition* when two or more tools are required: for instance, an agent booking a flight might need to consult a personal calendar for available times before filtering a list of flights. This requires an LLM to both break a task into smaller subtasks and select the correct tool for each subtask. Prior research

has shown that LLMs demonstrate limited capabilities in these situations in comparison to single-tool tasks (Qin et al., 2023a; Ye et al., 2025; Hosseini et al., 2024).

**Self-reflection for planning.** As LLMs display increasing reasoning capabilities, prior work has investigated self-reflection methods which allow LLMs to correct or critique their own responses in order to achieve better results (Madaan et al., 2023; Saunders et al., 2022; Tian et al., 2024a). Self-reflection has also been applied to tool use, as LLMs can use feedback from the tools (e.g. reasons for failure) to amend their outputs (Du et al., 2024; Gou et al., 2024; Qiao et al., 2023; Wang et al., 2024a). Using a multi-turn "chain-of-thought" allows a model to use its reasoning ability to better learn how to call tools (Chen et al., 2023b). This is particularly true when there are *silent* tool errors, where the tool fails without an error message but produces an incorrect or nonsensical output anyway. Sun et al. (2024) find that LLMs can frequently *detect* silent tool errors, for example in embodied settings with a object detector module, but often fail to recover from them, leading to cascading errors in multi-tool settings. In contrast, our emphasis is on the ability of LLMs to identify alternate solutions in large action spaces comprising large numbers of functions.

**LLM creativity.** Creative problem solving is often broken down into *convergent thinking*, or finding a single optimal solution to a problem, and *divergent thinking*, or coming up with multiple unique solutions to a problem (Ismayilzada et al., 2024; Lu et al., 2025). Divergent thinking is particularly important for adaptation to new constraints or environments; LLMs tend to struggle with both. Previous benchmarks for real-world creative thinking have evaluated on tasks such as escape-room puzzles (Qian et al., 2024) or real-world object usage (Tian et al., 2024b). While these benchmarks require creative thinking and compositionality of tools, their settings fall into the narrow category of physical tool use, preventing their use for assessment of virtual tools requiring API calls. Our benchmark examines performance with API requests. Aside from having a realistic setting, this has the advantage of inherent feedback mechanisms such as error message rather than relying on synthetic feedback in the form of human- or LLM-written descriptions of a fictional real-world setting.

# 3  Hell-or-High-Water: an Interactive Backup Planning Benchmark

## 3.1  Task Description

The general workflow for any given problem is as follows. The agent is presented with a description of the problem and a set of 4450 functions to potentially call. To navigate the large set of functions, we provide a semantic search tool to find and learn about available functions. If the agent uses the search tool by generating a query and executing the search tool, it will be provided with a retrieved set of functions and associated documentation relevant to the search query. The agent can then choose to either execute observed functions or look for other functions with a revised search query. If the agent chooses to try any of the observed functions, it must write code to execute those functions. After the code is executed, the agent observes the output and plans the next action accordingly. We provide a full trace in §F and more details regarding function search in §A.

In our benchmark, the agent interacts with our environment via the CodeAct framework (Wang et al., 2024b). CodeAct is an extension of ReAct (Yao et al., 2023), a prompting strategy where a language agent alternates between verbalized reasoning and taking actions on an environment. In CodeAct, the actions are concretely defined as executable blocks of code. The workflow is similar to that of using a Jupyter notebook, where a kernel executes one block of code at a time. By working in the modality of code, an LLM is theorized to better adhere to logical patterns of thought (Li et al., 2024). Thus, in our benchmark, each action an agent takes is expressed as a block of code, and each action is preceded by verbalized rationalization and planning. We view this workflow as blending traditional tool-use/reasoning approaches with code generation.

**Studying external errors.** So far we have outlined the agent's workflow under fault-free conditions. Recall that the agent must call the available Python functions to complete the task. If we throw errors during the agent's workflow, then we create external errors thereby inducing conditions requiring backup planning. By controlling when and how we

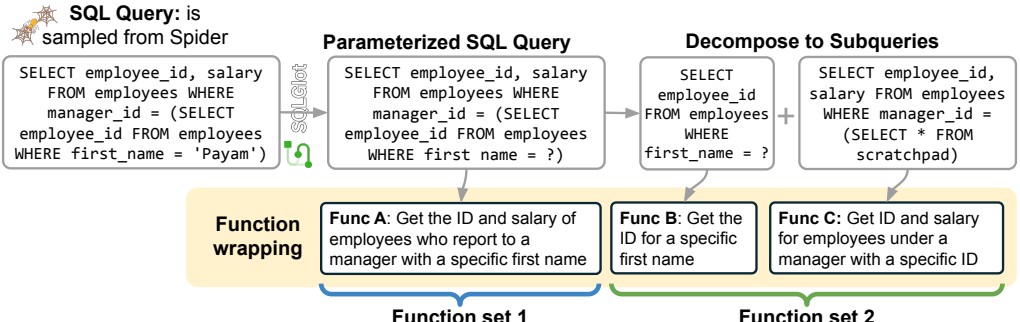

Figure 2: Benchmark creation pipeline. The "function sets" refer to groups of functions that together can be used to solve a given question: one direct solution involving a single function, and one composition of two or more functions that achieve the same result.

throw errors, we can study the effect of external errors on language agent performance. We describe this process in §A.

## 3.2 Benchmark Construction

We desire a method to automatically create functions with diverse functionality at scale. For this purpose, we turn to text-to-SQL datasets which pair natural language instructions with SQL queries that complete those instructions. In addition to already having instructions which we may reuse, datasets such as Spider (Yu et al., 2018) offer thousands of unique SQL queries. Additionally, SQL in these datasets can be queried against actual databases, allowing us to know the expected output for solution verification. Using the Spider dataset, we develop deterministic transformations of SQL to one or more function calls. This procedure results in a large set of functions with varying numbers of arguments and return types depending on the underlying SQL commands, which reflect the diverse functionality that a set of REST API endpoints might provide, all while ensuring functional correctness.

**Creating functions from SQL.** Using text-to-SQL datasets provides instructions that correspond to SQL queries. However, our benchmark calls for executable Python functions with inputs and outputs that an agent may use. At a high-level, we parameterize these SQL queries then wrap them in Python functions. We first parameterize the queries by substituting *literals*[3] with placeholder variables. This procedure is performed automatically by converting the SQL command into a parse tree[4], where each node represents an expression (Appendix C.1). To identify *literal expressions*, we perform depth first search on the parse tree. DFS yields the nodes of literal expressions in the order they appear-in in the SQL query, allowing us to replace those nodes with that of a placeholder variable.

By parameterizing the query, we deduce a set of parameters for it. If we wrap the query execution in a Python function, those parameters can serve as input arguments. The output of these functions consists of a list of JSON records. Each record is mapping of SQL identifiers to their corresponding value. In other words, the outputs are a subset of the rows and columns of a database, as specified by the SQL query it uses. We thus propose a pipeline for converting existing SQL queries into Python functions that can accept input and produce output. In §C.1 we walkthough an example of our function creation process.

**Function composition.** These problem types can be solved by a set of functions $\{f_1, ..., f_n\}$, where the input to $f_n$ is contained wholly in the output of $f_{n-1}$. Our benchmark creation process is able to produce problems of this nature. In cases where subqueries are present in the SQL query, we use our parser to locate and extract the subqueries. Each subquery is converted into tools in the same manner as before. Instructions in the Spider dataset that are mapped to SQL containing subqueries can be solved in at least two ways in our benchmark.

---

[3]i.e., values such as names or numbers that are explicitly expressed in the SQL query itself
[4]We use SQLGlot: https://github.com/tobymao/sqlglot

Our process of creating functions makes at least three when a subquery is present. Of these three functions, there exist two equivalent combinations of function calls—one involving the subqueries and the other involving the original parameterized query (§C.1).

**Function documentation.** Having created python functions out of SQL queries, we now prompt a language model[5] to create documentation for our tools. We adhere to the documentation format employed by OpenAI for their language model function-calling feature. This format is chosen because a growing number of language models (Dubey et al., 2024) have adopted it as their standard for code generation. Examples of documentation in §D.

**Quality control.** We removed questions that involved querying empty databases, or otherwise returned no results when the original Spider SQL query was run. Out of concern for context-length and the readability of function outputs, we removed questions where the original Spider SQL query returned an excess of 100 results. To ensure that every problem in our benchmark has a valid alternative solution, we use the intuition that SQL containing subqueries can be solved in at least two ways in our benchmark. We filter out all instructions in the Spider dataset that do not correpsond to a SQL query containing subqueries. After filtering, 413 questions remained. To increase the size of our benchmark, we augmented the remaining questions, which we describe in Appendix C. After augmentation, our benchmark includes 922 problems in total.

**Dataset statistics.** We portion a validation set of 92 examples to perform prompt engineering. This leaves a test set of 830 examples, each with at least 2 known combinations of function calls that yield a correct answer. Our benchmark creation process automatically creates a total of 4450 functions. Of these functions 2160 have no arguments, 1661 have one, 568 have two, 46 have 3, and 15 have 4.

# 4   Main Experiments

## 4.1   Experimental Setup

We provide empirical analysis of language model agent performance on recovering from external errors and identifying valid backup plans. Although there exist multiple ways to achieve the correct answer, we ensure that the agent encounters an external error (§A).[6]

**Models.**   We evaluate a wide range of LLMs in various sizes. Open weight models tested include Llama-3.1-Instruct, Llama-3.3-Instruct (Dubey et al., 2024), and Qwen-2.5-Instruct (Qwen et al., 2025). Proprietary models include Gemini-2.0-Flash and GPT-4o. All models were tested using greedy decoding for consistency. An analysis of the effect of model size on performance is in §5

**Evaluation procedure.** The models are evaluated by comparing their final prediction to the reference (human) answer, which is derived deterministically from the underlying SQL. Since the reference query produces the same output as a correct sequence of function calls, we can verify a solution's correctness simply by comparing it to the output of the reference query. In practice, an agent may provide a solution that is different from but functionally equivalent to the expected solution. For instance, the agent may choose to give the solution "John Doe" when the expected solution is [{"Name": "John Doe"}]. We handle these cases by applying deterministic post-processing to the final answers, in order to avoid unfairly penalizing models that do not strictly adhere to the expected type of the output. For each model we report the accuracy and standard error.[7]

---

[5]hugging-quants/Meta-Llama-3.1-70B-Instruct-AWQ-INT4

[6]Note that number of tool calls involved in the backup plan can vary. We show that this variable does not affect the difficulty of a problem

[7]Standard error here is derived from a bootstrapped confidence interval (0.95) of the accuracy using the `scipy` library

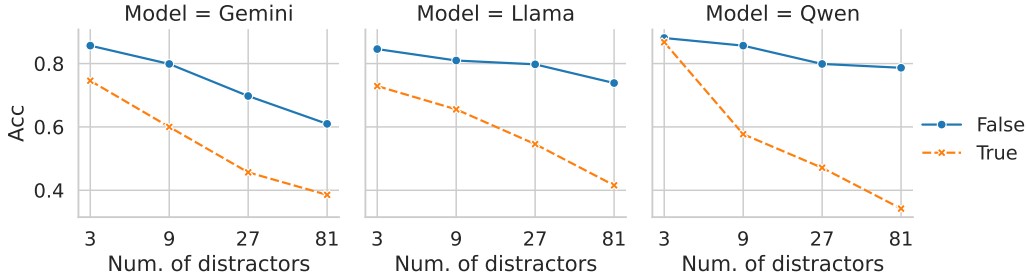

Figure 3: Closed world performance with a fixed set of functions provided in the context and no search tool. We show performance as the number of distractors is varied, both with (*True*) and without external errors (*False*). While fewer distractors corresponds to better performance, **the ability to form a backup plan in the face of external errors decreases as the size of the search space increases**.

## 4.2 Main Results

The results for the most difficult version of our benchmark, which requires language model agents to execute searches to find suitable functions from among thousands of candidates, are reported in Table 1.

| Model | External errors (N) | External errors (Y) | Acc. decrease |
|---|---|---|---|
| Gemini 2.0 (Flash) | $71.4 \pm 1.6$ | $41.1 \pm 1.7$ | 42.4% |
| GPT 4o | $60.5 \pm 1.7$ | $38.4 \pm 1.7$ | 36.5% |
| Llama 3.3 (70b, Instruct) | $64.0 \pm 1.7$ | $38.9 \pm 1.7$ | 39.2% |
| Llama 3.1 (70b, Instruct) | $42.3 \pm 1.7$ | $23.3 \pm 1.5$ | 44.9% |
| Qwen 2.5 (72B, Instruct) | $64.1 \pm 1.6$ | $35.3 \pm 1.7$ | 44.9% |

Table 1: Functional correctness (accuracy) in the open-world setting where language agents must use a search function to retrieve relevant tools. **All evaluated models exhibit a significant degradation in performance in the face of external errors.**

**Discussion.** The difference in performance between Llama-3.3 and Llama-3.1 is interesting, since these models are from the same family and perform quite similarly across certain common benchmark tasks such as MMLU. However, some of the largest improvements for Llama-3.3 are in instruction-following capabilities, coding and reasoning capabilities, achieving 92.1 on IFeval (Zhou et al., 2023), 88.4 on HumanEval (Chen et al., 2021), and 77.0 for MATH (Hendrycks et al., 2021) compared to 87.5, 80.5, and 67.8 respectively for Llama-3.1.[8] These improvements might partially explain the difference in performance on our benchmark, which requires robust instruction-following and reasoning capabilities.

**Closed world experiments.** Searching a large set of thousands of candidate functions introduces a significant challenge, on top of the challenge of forming backup plans. As an ablation of the impact of search, we experiment with a closed-world setting with a fixed number of functions that are provided upfront in the context. In these experiments, there is no search tool, and all functions necessary to address each question are provided, along with a varying number of *distractors*—extraneous functions not involved in the solution to the problem but provided in the prompt. These results are reported in Figure 3. Overall accuracy is significantly higher in this setting, but performance degrades to levels similar to those of the search setting (under 40% for $3^4$ distractors for Qwen2.5) in the face of external errors. We note that without distractors (when the number of functions is 3 in the $x$-axis), Qwen2.5 suffers only slight degradation from the introduction external errors, suggesting that the size of the search space plays an important role in the difficulty of our benchmark.

---

[8] https://www.llama.com/

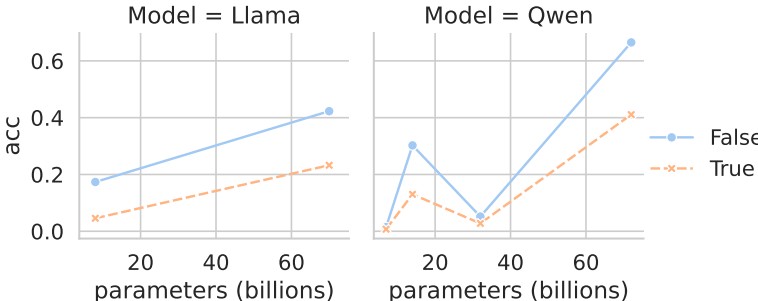

Figure 4: The effect of model size on performance, where "True" shows accuracy with forced function failures and "False" shows accuracy without function failures. **Agents consistently perform worse when external errors are introduced (dashed) regardless of model size.**

## 5 Further Analysis

**Error Analysis.** We analyze erroneous agent trajectories and categorize them with the following taxonomy—(1) *search failures*, where the correct tool never shows up in search results, (2) *ID failures*, where the correct tool shows up in search results, but the agent never uses it, (3) *chaining failures*, where the agent identifies one tool but not the other, or uses tools out of order, and (4) *tool use failures*, where the agent tries the correct tool, but gives up on using it correctly.

| Model | Total failures | Search | ID | Chaining | Tool use |
|---|---|---|---|---|---|
| Gemini 2.0 (Flash) | 489 | 57.1 | 25.5 | 12.3 | 6.14 |
| GPT 4o | 511 | 52.6 | 24.6 | 8.61 | 14.1 |
| Llama 3.3 (70b, Instruct) | 496 | 66.3 | 11.9 | 13.9 | 7.86 |
| Llama 3.1 (70b, Instruct) | 495 | 59.3 | 18.0 | 11.7 | 10.9 |
| Qwen 2.5 (72B, Instruct) | 637 | 60.5 | 26.4 | 7.06 | 6.12 |

Table 2: Breakdown of agent trajectory errors by type (in percent) for the results in Table 1. **Tool searching is the most common point of failure for all models tested.**

In the tool search setting, failure to find correct tools accounts for between 53% and 66% of the failures, followed by failure to select those tools when they appear in search results. Search errors may be explained in part by a failure to adequately compose queries reflecting the right information – a critical capability for agents interacting with large search spaces.

In the closed-world setting, to investigate the performance decrease as search space is increased, we apply our taxonomy to the Gemini-2 results. As the search space increases, the share of identification failures increases from 32.7% at 3 tools to 54.8% at 81 tools, suggesting that failure to recognize the correct tool(s) when provided is connected to this performance decrease. These results suggest that chaining tools together is less challenging than finding them in the first place.

| Tools | Total failures | ID | Chaining | Tool use |
|---|---|---|---|---|
| 3 | 211 | 32.7 | 16.1 | 51.2 |
| 9 | 332 | 45.8 | 23.8 | 30.4 |
| 27 | 451 | 54.8 | 16.9 | 28.4 |
| 81 | 510 | 54.9 | 15.5 | 29.6 |

Table 3: Breakdown of agent trajectory errors by type (in percent) for Gemini results in Figure 3. The percentage of tool searching errors increases with the total number of tools in the search space. **For large search spaces finding the right tools is more difficult than calling them in the right order**

**Does performance scale with size?**
We analyze the impact of model size on the ability to find a backup plan. Using the same setup as described in §4.1 we evaluate two sizes of LLama-3.1-Instruct and 4 sizes of Qwen-

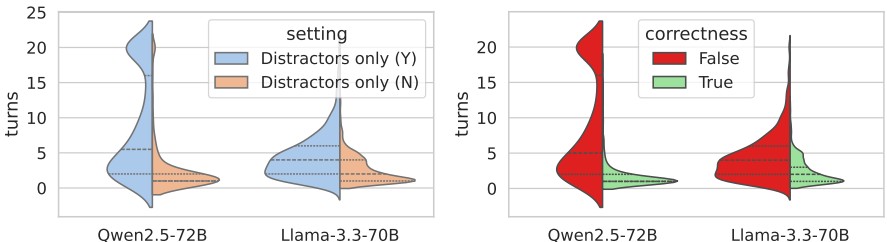

Figure 5: **(Left)** A violin plot showing how many turns the agent takes with only incorrect functions (Distractors only Y) and with correct functions included (Distractors only N). The two models exhibit markedly different performance in the Distractors only setting, with Qwen-2.5-Instruct particularly inclined to persist with impossible problems. **(Right)** Where correct functions *are* available, we plot how many turns the agent takes for correct solutions and incorrect solutions. In cases where the agent is correct, the agent uses fewer turns, suggesting that additional turns do not help the agent achieve the correct solution.

2.5-Instruct and show the results in Figure 4. We find that while absolute model performance tends to improve as model size increases, the model's resilience to failure does not, nor does this appear to be an emergent ability (Wei et al., 2022) of language models. Increasing model size does not address the performance degradation when agents face external errors, and other methods may be needed to give agents the ability to find and execute backup plans. For Qwen in particular, we notice a drop in performance for the 30B parameter model relative to the smaller model; examining the execution traces from this model, we notice that it has a tendency to ignore instructions regarding the formatting of responses, which is remedied with further scaling up to 70B.

**Do agents know when to give up?** So far, we have studied the behavior in situations where a correct solution is in principle achievable. However, in real-world settings, it may not be possible to satisfy a user request. In this experiment, we analyze the model's capability to recognize that it cannot solve a problem with the provided functions. We constrain the agent's set of possible functions such that it only has access to *incorrect* distractor functions, making the problem impossible to solve. Ideally, the agent should give up upon realizing this problem is impossible. We show the results for this setting in Figure 5. We find that this is not a clearly exhibited capability for all tested agents. The readiness of a model to give up varies by model; Llama-3.3-Instruct exhibits significantly more willingness to give up before exhausting the compute budget, while Qwen-2.5-Instruct generally fails to recognize that it cannot solve the task with the given functions. Figure 5 also shows the distribution of correct vs. incorrect answers for the default setting (where the correct functions are provided).

| Model | External errors (N) | External errors (Y) | Acc. decrease |
|---|---|---|---|
| Llama 3.3 (70b, Instruct) | $54.2 \pm 1.7$ | $36.5 \pm 1.7$ | 32.7% |
| Qwen 2.5 (72B, Instruct) | $71.1 \pm 1.6$ | $47.7 \pm 1.7$ | 32.9% |

Table 4: **What is the impact of underspecification?** This table shows results for two open-weight models for a modified version of the prompts where additional problem-specific hints are provided to reduce possible ambiguity (e.g., national elections vs. regional elections). Compared to §4.2, performance for Qwen is significantly improved, while Llama 3.3 actually performs worse. Nevertheless, we still observe a significant decrease in performance with the introduction of external errors.

**What if questions are underspecified?** In real-world settings, users may fail to provide sufficient information for a language agent to fulfill their request. For example, a user may ask a question about election results but fail to specify if they are asking about regional or national elections. In the context of our benchmark, we can control the degree of context included with each prompt to vary the amount of uncertainty for each question, and study the behavior of different LLMs under different degrees of uncertainty. The results from including additional disambiguating information to each question are provided in Table 4.

# 6 Discussion and Conclusion

**Main findings.** We find that language agents are not able to reliably incorporate feedback from the environment and identify backup plans. In the context of our benchmark, this is surprising for two main reasons. First, when a failure occurs, the feedback from the environment is quite explicit (e.g., "function_987 is currently unavailable. Please try a different function."). Second, our comparisons include LLMs which are specifically developed for applications such as the ones studied in this paper. For example, Google Gemini 2.0 features "action-capabilities, [...], long context understanding, complex instruction following and planning, compositional function-calling,"[9] but, while it performs best among the LLMs we compare—performing 11 points better than GPT-4o in the *no failure* setting—Gemini 2.0 still has a sharp drop in accuracy when forced to explore alternate solutions. Model scale also does seem to matter; in the category of open-weight models, the 70B parameter configurations perform drastically better than smaller ones.

**Future work.** Why are language agents unable to adapt effectively to unexpected feedback? Our analysis uncovers some common issues among LLMs (Jiang et al., 2025). Since our benchmark involves new functions, the biases of the language agents may be quite harmful due to the need to adapt to novel situations. For example, Figure 5 shows markedly different behavior for two otherwise similar models, which may suggest that training biases are overriding the behavior of the model, even when faced with clear-cut feedback from the environment. It would also be interesting to explore whether recent thinking/reasoning models display improved fault-tolerance. Our preliminary experiments with thinking models suggested that they may require different prompting strategies; however, their increased cost precluded a more detailed analysis.

**Memory.** We highlight that our context window itself serves as a form of episodic memory for the agent, in which past attempts, mistakes, and reflections are kept in-context over the course of solving a problem for the benefit of the agent. However, agentic workflows can incorporate memory that persists across episodes for the purpose of lifelong learning (Shinn et al., 2023). While we recognize that persistent memory and lifelong learning are an important next-step in the improvement of agentic capabilities, the purpose of our benchmark is to enable research in improving backup planning. Therefore we leave exploration of persistent memory to future research.

**Limitations.** We do not evaluate the latest commercial models due to cost. For dataset creation, our work builds on the Spider text-to-SQL benchmark (Yu et al., 2018). Although Spider has been relatively widely used and has undergone improvements and revisions, we find that it is imperfect and contains instances of under-specified questions, as well as data quality issues such as empty tables, which we had to manually filter out. Nonetheless, by basing our benchmark on human-generated source data, we can verify if generated answers match the reference without being prescriptive about the solution, which is important for our research questions. As previously mentioned, we find that the studied LLMs can be sensitive to the prompt specification. Therefore, we use an established prompting framework based on ReAct, but we acknowledge that further LLM-specific prompt design could improve the performance of specific LLMs. Additionally, most SOTA models are trained to use specific prompt templates for tool-calling. However, these templates are not standardized across models. To control for these differences, and to make implementation tractable, we use the default instruction following capability of these models instead of custom tool-calling functionalities. We hope that our work motivate future work in improving the fault-tolerance of LLMs, via prompt design or more fundamental training changes.

## Acknowledgments

We would like to thank Rachel Wicks, Cristina Aggazzotti, Rafael Rivera-Soto, Aleem Khan, Ashi Garg, Jack Zhang, Ricky Mouser, and Jiefu Ou as well as our anonymous reviewers for their spirited discussion and feedback.

---

[9]Gemini 2.0 blog post

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

# A  Additional Task Information

We here provide further context into the task definition discussed in §3.1.

**Function search.**  To allow an agent to explore what tools are available in our environment, we provide two special functions. The output of each function is observed and processed by the agent in accordance with CodeAct. The first function, search_tools, accepts a search query as input and returns a list of relevant tool names with a brief description. The number of tools returned can be specified by the agent. Should the agent wish to learn more about a tool, such as the names of arguments, it can call the second function get_info. This function accepts a tool name as input and returns the full documentation for the tool, which includes a list of required arguments and a description of their function. We design our search functions in this way in the hopes of reducing the number of tokens an LLM must process in context: documentation can contain many tokens, and therefore it should only be returned if necessary.

**Studying external errors (continued).**  For a given problem we are given knowledge of $F_s$, or the set of all functions that can lead to a correct solution. We can replace the first function $f \in F_s$ that the agent tries with a modified version $f'$ that always throws an error. With the function $f$ effectively unusable, the agent must discover and compose the other functions in $F_s$ over the course of solving the problem.[10] We therefore view our method of introducing inexplicable function errors as a means to study the ability of language agents to find a successful backup plan when their original plan fails unexpectedly.

# B  Additional Analyses

| Model | Single Call | Multi Call | Ratio (Single:Multi) |
|---|---|---|---|
| Gemini 2.0 (Flash) | 38.3 | 43.2 | 355:472 |
| GPT 4o | 39.9 | 37.3 | 358:472 |
| Llama 3.3 (70b, Instruct) | 30.0 | 49.5 | 390:400 |
| Llama 3.1 (70b, Instruct) | 14.5 | 40.0 | 545:285 |
| Qwen 2.5 (72B, Instruct) | 38.9 | 41.8 | 445:385 |

Table 5: Accuracy stratified by number of tool calls in backup plan for tool search results in Table 1. **Backup plans involving multiple tools are not substantially more challenging than those involving a single tool**

**Are backup plans involving multiple tools more challenging to identify than those involving a single tool?**  For the results in Table 1, we stratify accuracy based on the number of function calls involved in the available solution, and find that, if anything, single call solutions were more challenging for a few models (Llama). Otherwise, we did not observe much performance asymmetry. In addition to the accuracy, we provide the ratio of single function calls to multi function calls and show that ratio is fairly balanced.

**Are different models getting the same questions wrong?**  For the results in Table 1, we calculated Fleiss's Kappa to see whether different models "agreed" on the incorrect questions. Our results indicate an agreement of .246, showing "fair" but somewhat low agreement. This suggests that different models tend to get different questions wrong. In other words, model behavior rather than inherent data uncertainty accounts for our results.

# C  Additional Benchmark Creation Details

We provide further context into the benchmark construction discussed in §3.2.

---

[10]Recall that in practice, there will always exist multiple subsets of $F_s$ that lead to a correct solution

**Function documentation (continued)**   We provide the language model the code for the function itself, a brief description of the database it accesses, and two in-context demonstrations. To ensure the quality of our synthetic documentation, we pass them through a series of unit tests, which include checks for the presence of important fields and alignment with code. If the generated documentation fails to pass the checks, the language model has three attempts to correct itself. In practice, we find that documentation which passes the unit tests is produced for all tools we create.

**Names of functions and arguments.**   LLMs train on large amounts of code during pre-training. The effects of code seen during pre-training can confound the conclusions of our benchmark—the agent may rely on memorization of prior examples to complete the task, rather than genuine reasoning. To control for the effects of previously-seen code in our benchmark, we give un-descriptive names to tools and their input argument. This way, the purpose of a tool cannot be guessed through its name or arguments, forcing the agent to consult documentation.

**Function composition (continued).**   While the process of creating Python functions from SQL subqueries may appear straightforward, deducing the arguments for a function $f_n$ based on the outputs of the previous tool $f_{n-1}$ is not. Tool outputs take the form of a list of JSONs, but tool inputs in our benchmark take the form of SQL literals. To reconcile this difference, we extract values from the JSON list by field name, creating a mapping field name (str) → values (list). This allows us to set the input arguments for a tool $T_n$ to the fields of the previous $f_{n-1}$, where the expected input for each argument is a list of values. This introduces another issue—inputs are Pythonic datatypes, but the underlying functionality is in SQL. To reconcile the input values of Pythonic datatypes with the wrapped SQL query, we create a temporary SQL table called scratchpad that loads the input values to the function. This and other implementation details can be found in an example of a Python function below.

**Data augmentation.**   Consider the instruction "What are the employee ids of employees who report to Shelley, and what are their salaries?" This corresponds to the paramterized query SELECT employee_id, salary FROM employees WHERE manager_id = (SELECT employee_id FROM employees WHERE **first_name = ?**). As a byproduct of our automatic function creation pipeline, we know for certain instructions, the input values for calling the appropriate function are present verbatim in the text (e.g., "Shelley"). We can use this knowledge to alter the value that appears in the instruction with other values present in the same column of the database. We can also parameterize the query with these alternative values and run them against the database to obtain the updated reference solutions. As a result we are able to more than double the size of our benchmark.

### C.1   Function Creation Example

We provide an example for creating executable python functions from SQL commands (discussed in §3.2).

**1. SQL**   We begin the process of automated Python function creation with a SQL query.

```
SELECT employee_id, salary FROM employees WHERE manager_id = (SELECT employee_id FROM employees WHERE
    first_name = 'Payam')
```

**2. Parsed SQL**   We convert the SQL command into a parse tree using SQLGlot (Figure 3.2), where each node represents an expression:

```
Select(
  expressions=[
    Column(
      this=Identifier(this=employee_id, quoted=False)),
    Column(
      this=Identifier(this=salary, quoted=False))],
  from=From(
    this=Table(
```

```
              this=Identifier(this=employees, quoted=False))),
    where=Where(
      this=EQ(
        this=Column(
          this=Identifier(this=manager_id, quoted=False)),
        expression=Subquery(
          this=Select(
            expressions=[
              Column(
                this=Identifier(this=employee_id, quoted=False))],
            from=From(
              this=Table(
                this=Identifier(this=employees, quoted=False))),
            where=Where(
              this=EQ(
                this=Column(
                  this=Identifier(this=first_name, quoted=False)),
                expression=Literal(this=Payam, is_string=True))))))))))
```

**3. Query Parametrization**    We parameterize the query by substituting literals with placeholder variables. This is done deterministically with a depth-first-search for `literal` type expressions.

```
Select(
  expressions=[
    Column(
      this=Identifier(this=employee_id, quoted=False)),
    Column(
      this=Identifier(this=salary, quoted=False))],
  from=From(
    this=Table(
      this=Identifier(this=employees, quoted=False))),
  where=Where(
    this=EQ(
      this=Column(
        this=Identifier(this=manager_id, quoted=False)),
      expression=Subquery(
        this=Select(
          expressions=[
            Column(
              this=Identifier(this=employee_id, quoted=False))],
          from=From(
            this=Table(
              this=Identifier(this=employees, quoted=False))),
          where=Where(
            this=EQ(
              this=Column(
                this=Identifier(this=first_name, quoted=False)),
              expression=Placeholder())))))))
```

We use SQLGlot to convert the modified parse tree back to the following parametrized query:

```
SELECT employee_id, salary FROM employees WHERE manager_id = (SELECT employee_id FROM employees WHERE
    first_name = ?)
```

**4. Subquery Decomposition**    In cases where subqueries are present in the SQL query, we use our parser to locate and extract the subqueries. Considered in conjunction with the parametrized query from the previous step, this provides 3 synthesized queries.

```
SELECT employee_id FROM employees WHERE first_name = ?
```

```
SELECT employee_id, salary FROM employees WHERE manager_id = (SELECT * FROM scratchpad)
```

**5. Function Wrapping**    Having parametrized the queries and identified the potential input arguments, we wrap the queries in a Python function. Note that a SQL query with a single subquery can be made into 3 functions (function_985, function_986, and function_987). Of these functions, calling function_987 is equivalent to calling function_985 followed by function_986, assuming correct input arguments.

```
def function_985(lambda_sigma: str):
    ### SETUP ###
```

```python
    # connect to database
    db_path = "spider_data/database/hr_1/hr_1.sqlite"
    conn = sqlite3.connect(db_path)
    cursor = conn.cursor()

    ### EXECUTE QUERY ###
    query = "SELECT employee_id FROM employees WHERE first_name = ?"
    args = (lambda_sigma,)
    cursor.execute(query, args)
    res = cursor.fetchall()
    columns = [description[0] for description in cursor.description]
    data = [dict(zip(columns, row)) for row in res]
    cursor.close()
    conn.close()

    ### HINT ###
    if not len(data) or not len([_ for _ in res[0] if _ is not None]):
        from difflib import get_close_matches
        var_identifiers = {"lambda_sigma": "first_name"}
        var_aliases = ["lambda_sigma"]
        var_types = {"lambda_sigma": "str"}
        var_data = [lambda_sigma]
        var_values = dict(zip(var_aliases, var_data))
        with open("lookup_table_cache.json", 'r') as infile:
            column_data_lookup = json.load(infile)
        for alias, identifier_ in var_identifiers.items():
            hint_data = column_data_lookup['hr_1'][identifier_]['values']
            arg_value = var_values[alias]
            if var_types[alias] == 'str' and arg_value not in hint_data:
                hint_data = {string.lower(): string for string in hint_data}
                print(f"Could not find entry with `{alias}` value '{arg_value}'")
                close_matches = get_close_matches(arg_value.lower(), list(hint_data.keys()), n=1, cutoff
   =.8)

                if close_matches:
                    print(f"Did you mean '{hint_data[close_matches[0]]}' ?")

    return data
```

```python
def function_986(beta_epsilon: list):
    ### PREPROCESSING ###
    # organize input data as list
    input_data = [beta_epsilon]
    # map identifier aliases to names
    identifier_info = {"EMPLOYEE_ID": "beta_epsilon"}
    identifier_types = {"beta_epsilon": "list"}
    # map identifier names to input data
    result_ = dict(zip(identifier_info.keys(), [d if type(d) == list else [d] for d in input_data]))
    for identifier_alias, value in zip(identifier_info.values(), input_data):
        assert identifier_types[identifier_alias] == str(type(value).__name__), f"Expected {
  identifier_types[identifier_alias]} but got {type(value)} for {identifier_alias}"
    # transpose result_ into result
    list_length = len(next(iter(result_.values())))
    result = []
    for i in range(list_length):
        temp_record = {}
        for identifier, values in result_.items():
            temp_record[identifier] = values[i]
        result.append(temp_record)

    ### SETUP ###
    # connect to database
    db_path = "spider_data/database/hr_1/hr_1.sqlite"
    conn = sqlite3.connect(db_path)
    cursor = conn.cursor()

    # create scratchpad
    quotes = "\""
    cursor.execute('DROP TABLE IF EXISTS scratchpad') # remove scratchpad if it exists
    temp_query = f"CREATE TEMPORARY TABLE scratchpad ([{', '.join([k for k, _ in result_.items()])}])"
    cursor.execute(temp_query)
    for temp_record in result:
        temp_insert = f'INSERT INTO scratchpad ([{", ".join([k for k, _ in result_.items()])}]) VALUES
  ({", ".join([quotes + str(val) + quotes if type(val) == str else str(val) for val in temp_record.
  values()])});'
        cursor.execute(temp_insert)

    ### EXECUTE QUERY ###
    query = "SELECT employee_id, salary FROM employees WHERE manager_id = (SELECT * FROM scratchpad)"
    cursor.execute(query)
    res = cursor.fetchall()
    columns = [description[0] for description in cursor.description]
    data = [dict(zip(columns, row)) for row in res]
    cursor.close()
    conn.close()
```

```
    ### HINT ###
    if not len(data) or not len([_ for _ in res[0] if _ is not None]):
        from difflib import get_close_matches
        var_identifiers = {}
        var_aliases = []
        var_types = {}
        var_data = []
        var_values = dict(zip(var_aliases, var_data))
        with open("lookup_table_cache.json", 'r') as infile:
            column_data_lookup = json.load(infile)
        for alias, identifier_ in var_identifiers.items():
            hint_data = column_data_lookup['hr_1'][identifier_]['values']
            arg_value = var_values[alias]
            if var_types[alias] == 'str' and arg_value not in hint_data:
                hint_data = {string.lower(): string for string in hint_data}
                print(f"Could not find entry with `{alias}` value '{arg_value}'")
                close_matches = get_close_matches(arg_value.lower(), list(hint_data.keys()), n=1, cutoff
 =.8)
                if close_matches:
                    print(f"Did you mean '{hint_data[close_matches[0]]}' ?")

    return data
```

```
def function_987(lambda_epsilon: str):
    ### SETUP ###
    # connect to database
    db_path = "spider_data/database/hr_1/hr_1.sqlite"
    conn = sqlite3.connect(db_path)
    cursor = conn.cursor()

    ### EXECUTE QUERY ###
    query = "SELECT employee_id, salary FROM employees WHERE manager_id = (SELECT employee_id FROM
      employees WHERE first_name = ?)"
    args = (lambda_epsilon,)
    cursor.execute(query, args)
    res = cursor.fetchall()
    columns = [description[0] for description in cursor.description]
    data = [dict(zip(columns, row)) for row in res]
    cursor.close()
    conn.close()

    ### HINT ###
    if not len(data) or not len([_ for _ in res[0] if _ is not None]):
        from difflib import get_close_matches
        var_identifiers = {"lambda_epsilon": "first_name"}
        var_aliases = ["lambda_epsilon"]
        var_types = {"lambda_epsilon": "str"}
        var_data = [lambda_epsilon]
        var_values = dict(zip(var_aliases, var_data))
        with open("lookup_table_cache.json", 'r') as infile:
            column_data_lookup = json.load(infile)
        for alias, identifier_ in var_identifiers.items():
            hint_data = column_data_lookup['hr_1'][identifier_]['values']
            arg_value = var_values[alias]
            if var_types[alias] == 'str' and arg_value not in hint_data:
                hint_data = {string.lower(): string for string in hint_data}
                print(f"Could not find entry with `{alias}` value '{arg_value}'")
                close_matches = get_close_matches(arg_value.lower(), list(hint_data.keys()), n=1, cutoff
 =.8)
                if close_matches:
                    print(f"Did you mean '{hint_data[close_matches[0]]}' ?")

    return data
```

## D   Examples of documentation

The following are examples of documentation produced automatically via LLM prompting.

```
function_985: {
    "type": "function",
    "function": {
        "name": "function_985",
        "description": "Get the employee_id from the `employees` table for a specific first_name. The
    employees table stores information about employees in a multinational corporation's human resources
    management system",
        "parameters": {
            "type": "object",
            "properties": {
                "lambda_sigma": {
                    "type": "str",
                    "description": "The first_name to query"
```

```
                }
            },
            "required": [
                "lambda_sigma"
            ]
        }
    }
}
function_986: {
    "type": "function",
    "function": {
        "name": "function_986",
        "description": "Get the employee_id and salary from the `hr_1` database for employees with a
    manager_id matching the provided input. The `hr_1` database is a human resources management system
    storing information about employees, departments, job roles, locations, and countries, as well as
    employment history and organizational structure.",
        "parameters": {
            "type": "object",
            "properties": {
                "beta_epsilon": {
                    "type": "list",
                    "description": "A list of manager_ids to query"
                }
            },
            "required": [
                "beta_epsilon"
            ]
        }
    }
}
function_987: {
    "type": "function",
    "function": {
        "name": "function_987",
        "description": "Get the employee_id and salary of employees who report to a manager with a
    specific first name. The data is retrieved from a human resources management system database.",
        "parameters": {
            "type": "object",
            "properties": {
                "lambda_epsilon": {
                    "type": "str",
                    "description": "The first name of the manager to query"
                }
            },
            "required": [
                "lambda_epsilon"
            ]
        }
    }
}
```

We include the documentation used for search functions.

```
search_tools: {
    "type": "function",
    "function": {
        "name": "search_tools",
        "description": "Retrieves descriptions of tools that are relevant to a provided search query.
    This function can be used to explore what tools are available to use.",
        "parameters": {
            "type": "object",
            "properties": {
                "query": {
                    "type": "string",
                    "description": "description used to search for tools"
                },
                "num_results": {
                    "type": "integer",
                    "description": "number of results to return (max 9)",
                    "default": 9
                }
            },
            "required": ["search_query"]
        }
    }
}
get_info: {
    "type": "function",
    "function": {
        "name": "get_info",
        "description": "Retrieves the documentation for a specific tool. The documentation will include
    the tool's name, description, and usage instructions.",
        "parameters": {
            "type": "object",
```

```
            "properties": {
                "tool_name": {
                    "type": "string",
                    "description": "name of the tool to retrieve documentation for"
                }
            },
            "required": ["tool_name"]
        }
    }
}
```

# E   Prompts

We here provide the prompts we use, both for in-context functions and for function search.

## E.1   In-context functions

```
You are a helpful assistant assigned with the task of problem-solving. To achieve this, you will be using
    an interactive coding environment equipped with a variety of tool functions to assist you
    throughout the process.

At each turn, you should first provide your step-by-step thinking for solving the task. Your thought
    process should be enclosed using "<thought>" tag, for example: <thought> I need to print "Hello
    World!" </thought>.

After that, you have two options:

1) Interact with a Python programming environment and receive the corresponding output. Your code should
    be enclosed with the "<execute>" tag, for example: <execute> print("Hello World!") </execute>.
2) Stop execution when you have encountered a correct solution. Save the final answer to the `solution`
    variable and enclose the code with the "<solution>" tag, for example: <solution> solution = output
    </solution>.

To help us verify your solution, your solution should contain the values and *original keys* returned by
    the tools

Each tool will return a list of JSONs, and you may need to perform postprocessing on the output. Print
    the output to see how the data is structured.

Here are some examples (tool definitions in each example have been truncated for brevity):

---

{{
    tool definitions omitted...
}}
...

Task: What is the names of movies whose created year is after all movies directed by Steven Spielberg?

Assistant:
<thought> I can use function_3694 to get the movies released after Steven Spielberg's last movie </
    thought>

<execute>
output = function_3694('Steven Spielberg')
result = [item['movie'] for item in output]
print(result)
</execute>

Observation:
Traceback (most recent call last):

  Cell In[1], line 2
    result = [item['movie'] for item in output]

  Cell In[1], line 2 in <listcomp>
    result = [item['movie'] for item in output]

KeyError: 'movie'

Assistant:
<thought> 'movie' doesn't appear to be a key in the outputs. To correct the error, I will first try
    printing out the output of function_3694 </thought>

<execute>
output = function_3694('Steven Spielberg')
print(output)
</execute>
```

```
Observation:
[{{'title': 'Titanic'}}, {{'title': 'Avatar'}}]

Assistant:
<thought> The output seems reasonable. The movies released after Steven Spielberg's last movie are '
      Titanic' and 'Avatar'. I will report the solution below </thought>

<solution>
solution = output
</solution>

---

{{
    tool definitions omitted...
}}
...

Task: Find the average age of students who are living in the dorm with the largest capacity.

Assistant:
<thought> I need to use function_3643 to figure out the largest capacity, then use function_3644 to get
      the average of the students living in the dorm with that capacity. I will first run function_3643 </
      thought>

<execute>
output = function_3643()
print(output)
</execute>

Observation:
[{{'MAX(student_capacity)': 400}}]

Assistant:
<thought> Now I will postprocess the output of function_3643 to use as the input to function_3644 </
      thought>

<execute>
input1 = output[0]['MAX(student_capacity)']
output2 = function_3644(input1)
print(output2)
</execute>

Observation:
Traceback (most recent call last):

  Cell In[1], line 2
    output2 = function_3644(input1)

  File ~/sandbox_utils.py:21114 in function_3644
    assert field_types[field_name] == str(type(value).__name__), f"Expected {{field_types[field_name]}}
      but got {{type(value)}} for {{field_name}}"

AssertionError: Expected float but got <class 'int'> for xray_tau

Assistant:
<thought> It seems my code had an error. Based on the error, I will convert the input from an int to a
      float </thought>

<execute>
input1 = output[0]['MAX(student_capacity)']
output2 = function_3644(float(input1))
print(output2)
</execute>

Observation:
[{{'AVG(T1.age)': 19.6}}]

Assistant:
<thought> The result seems reasonable. The average age of the students in the dorm with the largest
      capacity is 19.6. I will report the solution below </thought>

<solution>
solution = output2
</solution>

---

Here are some of the tools you can use for the current task:
[FUNCTIONS HERE]
```

## E.2   Function search

```
You are a helpful assistant assigned with the task of problem-solving. To achieve this, you will be using
    an interactive coding environment equipped with a variety of tool functions to assist you
    throughout the process.

At each turn, you should first provide your step-by-step thinking for solving the task. Your thought
    process should be enclosed using "<thought>" tag, for example: <thought> I need to print "Hello
    World!" </thought>.

After that, you have two options:

1) Interact with a Python programming environment and receive the corresponding output. Your code should
    be enclosed with the "<execute>" tag, for example: <execute> print("Hello World!") </execute>.
2) Stop execution when you have encountered a correct solution. Save the final answer to the `solution`
    variable and enclose the code with the "<solution>" tag, for example: <solution> solution = output
    </solution>.

To help us verify your solution, your solution should contain the values and *original keys* returned by
    the tools

Each tool will return a list of JSONs, and you may need to perform postprocessing on the output. Print
    the output to see how the data is structured.

Here are some examples (tool definitions in each example have been truncated for brevity):

---

{{
    tool definitions omitted...
}}
...

Task: What is the names of movies whose created year is after all movies directed by Steven Spielberg?

Assistant:
<thought> To find the names of movies created after all movies directed by Steven Spielberg, I first need
    to find the year of Steven Spielberg's last movie and then find all the movies that were released
    after. I can use the search_tools function to find relevant tools that can help with this task. </
    thought>

<execute>
search_tools(query='movie directors', num_results=9)
</execute>

Observation:
['{{"function_4359": "Get the number of distinct movie titles directed by a specific director. The imdb
    database stores information about movies and TV series, including their casts, crews, genres,
    keywords, and production details."}}',
 '{{"function_4161": "Get the directors from the `movie_1` database who have directed a specific number of
    movies. The movie_1 database stores information about movies, reviewers, and their corresponding
    ratings."}}',
 '{{"function_74": "Retrieve a list of film titles and their corresponding directors from the cinema
    database."}}',
 '{{"function_2309": "Get a list of distinct directors from the `film_rank` database. The film_rank
    database stores information about films, including their titles, studios, directors, and box office
    gross, as well as market estimations for these films in various countries."}}',
 '{{"function_3429": "Get the count of distinct movie titles directed by a specific director and released
    before a certain year. The imdb database stores information about movies and TV series, including
    their casts, crews, genres, keywords, and production details."}}'
 ...results truncated]

Assistant:
<thought> These tools don't seem like they would help us solve the task. I should refine my search query
    to be more specific and see if that returns more relevant tools </thought>

<execute>
search_tools(query='movie directors year', num_results=9)
</execute>

Observation:
['{{"function_3648": "Get the count of distinct movie titles directed by a specific director, released
    within a specified year range. The imdb database stores information about movies and TV series,
    including their casts, crews, genres, keywords, and production details."}}',
 '{{"function_3429": "Get the count of distinct movie titles directed by a specific director and released
    before a certain year. The imdb database stores information about movies and TV series, including
    their casts, crews, genres, keywords, and production details."}}',
 '{{"function_1933": "Get the count of distinct movie titles directed by a specific director and released
    after a certain year. The imdb database stores information about movies and TV series, including
    their casts, crews, genres, keywords, and production details."}}',
 '{{"function_1451": "Get the distinct directors of films released in a specific year. The function uses
    data from the `film_rank` database, which stores information about films and their market
    estimations."}}',
 '{{"function_3692": "Get the latest year of release for movies directed by a specific director. The data
    is sourced from the `movie_1` database, which stores information about movies, reviewers, and their
    corresponding ratings."}}'
 ...results truncated]
```

```
Assistant:
<thought> I can use function_3692 to find the year of Steven Spielberg's last movie. I should find out
    more about function_3692 using the get_info tool </thought>

<execute>
get_info('function_3692')
</execute>

Observation:
{{
    tool definitions omitted...
}}

Assistant:
<thought> I can call function_3692 with the lambda_mu parameter to get the year of Steven Spielberg's
    last movie </thought>

<execute>
year = function_3692(lambda_mu='Steven Spielberg')
print(year)
</execute>

Observation:
[{{'MAX(YEAR)': 1982}}]

Assistant:
<thought> Steven Spielberg's last movie was released in 1982. I need to find all the movies that were
    released after. I can use the search_tools function to find relevant tools that can help with this
    task </thought>

<execute>
search_tools('movie after year', num_results=9)
</execute>

Observation:
['{{"function_3693": "Get the titles of movies released after a specified year. The movie_1 database
    stores information about movies, reviewers, and their corresponding ratings, allowing for the
    tracking and analysis of movie reviews and ratings from multiple reviewers."}}',
 '{{"function_3628": "Get the number of unique movie titles released in a specific year. The imdb
    database stores information about movies and TV series, including their casts, crews, genres,
    keywords, and production details."}}',
 '{{"function_1933": "Get the count of distinct movie titles directed by a specific director and released
    after a certain year. The imdb database stores information about movies and TV series, including
    their casts, crews, genres, keywords, and production details."}}',
 '{{"function_1976": "Get movie titles from the `imdb` database for a specific genre and release year. The
    imdb database stores information about movies and TV series, including their casts, crews, genres,
    keywords, and production details."}}',
 '{{"function_3648": "Get the count of distinct movie titles directed by a specific director, released
    within a specified year range. The imdb database stores information about movies and TV series,
    including their casts, crews, genres, keywords, and production details."}}'
 ...results truncated]

Assistant:
<thought> I can use function_3693 to find the titles of movies released after 1982. I should find out
    more about function_3693 using the get_info tool </thought>

<execute>
get_info(tool_name='function_3693')
</execute>

Observation:
{{
    tool documentation omitted...
}}

Assistant:
<thought> I can call function_3693 with the beta_papa parameter to get the titles of movies released after
    1982 </thought>

<execute>
titles = function_3693(beta_papa=year[0]['MAX(YEAR)'])
print(titles)
</execute>

Observation:
Traceback (most recent call last):

  Cell In[1], line 1
    titles = function_3693(beta_papa=year[0]['MAX(YEAR)'])

  File ~/sandbox_utils.py:159922 in function_3693
    assert identifier_types[identifier_alias] == str(type(value).__name__), f"Expected {{identifier_types
    [identifier_alias]}} but got {{type(value)}} for {{identifier_alias}}"

AssertionError: Expected float but got <class 'int'> for beta_papa
```

```
Assistant:
<thought> It seems my code had an error. Based on the error, I will convert the input from an int to a
    float </thought>

<execute>
titles = function_3693(beta_papa=float(year[0]['MAX(YEAR)']))
print(titles)
</execute>

Observation:
[{{'title': 'Titanic'}}, {{'title': 'Avatar'}}]

<thought> The output seems reasonable. The movies released after Steven Spielberg's last movie are '
    Titanic' and 'Avatar'. I will report the solution below </thought>

<solution>
solution = titles
</solution>

---

Task: Find the average age of students who are living in the dorm with the largest capacity.

Assistant:
<thought> To find the average age of students in the dorm with the largest capacity, I first need to find
    the dorm with the largest capcity and then find the average age of students living there. I can use
    the search_tools function to find relevant tools that can help with this task. </thought>

<execute>
search_tools(query="age students dorm capacity", num_results=9)
</execute>

Observation:
['{{"function_3644": "Get the average age of students living in dorms with a specific student capacity.
    The function uses data from the `dorm_1` database, which manages information about university
    students, including their personal details, dorm assignments, and the amenities available in each
    dorm."}}',
 '{{"function_3643": "Get the maximum student capacity from the dorm information. The dorm information is
    managed by the `dorm_1` database, which stores data about university students, including their
    personal details, dorm assignments, and the amenities available in each dorm."}}',
 '{{"function_2491": "Get the total student capacity across all dorms. The data is retrieved from the `
    dorm_1` database, which manages information about university students, including their personal
    details, dorm assignments, and the amenities available in each dorm."}}',
 '{{"function_3645": "Get the average age of students living in the dorm with the highest student capacity
    . The function retrieves data from the `dorm_1` database, which manages information about university
     students, including their personal details, dorm assignments, and the amenities available in each
    dorm."}}',
 '{{"function_2676": "Get the count of students and dorm names for students older than a specified age.
    The data is retrieved from the `dorm_1` database, which manages information about university
    students, including their personal details, dorm assignments, and the amenities available in each
    dorm."}}'
 ...results truncated]

<thought> I can use function_3645 to directly find the average age of students living in the dorm with
    highest capacity. I should find out more about function_3645 using the get_info tool </thought>

<execute>
get_info(tool_name="function_3645")
</execute>

Observation:
{{
    tool documentation omitted...
}}

<thought> I can call function_3645 to find the average age of students living in the dorm with highest
    capacity </thought>

<execute>
average_age = function_3645()
print(average_age)
</execute>

Observation:
[{{'AVG(T1.age)': 19.6}}]

<thought> The result seems reasonable. The average age of the students in the dorm with the largest
    capacity is 19.6. I will report the solution below </thought>

<solution>
solution = average_age
</solution>

---
```

```
To start with, you will have access to two tools to help you explore other tools that are available to
    use.

search_tools = {
    "type": "function",
    "function": {
        "name": "search_tools",
        "description": "Retrieves descriptions of tools that are relevant to a provided search query.
    This function can be used to explore what tools are available to use.",
        "parameters": {
            "type": "object",
            "properties": {
                "query": {
                    "type": "string",
                    "description": "description used to search for tools"
                },
                "num_results": {
                    "type": "integer",
                    "description": "number of results to return (max 9)",
                    "default": 9
                }
            },
            "required": ["search_query"]
        }
    }
}

get_info = {
    "type": "function",
    "function": {
        "name": "get_info",
        "description": "Retrieves the documentation for a specific tool. The documentation will include
    the tool's name, description, and usage instructions.",
        "parameters": {
            "type": "object",
            "properties": {
                "tool_name": {
                    "type": "string",
                    "description": "name of the tool to retrieve documentation for"
                }
            },
            "required": ["tool_name"]
        }
    }
}
```

## F   Example Trace (Success)

We include an example of successful task completion with Qwen2.5-72B-Instruct.

```
Task: What are the employee ids of employees who report to Shelley, and what are their salaries?
<thought> To find the employee ids and salaries of employees who report to Shelley, I need to find a tool
    that can query the employee database and filter by manager name. I can use the search_tools
    function to find relevant tools. </thought>

<execute>
search_tools(query="employee manager salary", num_results=9)
</execute>

Observation:
['{"function_1800": "Get the salary of an employee from the `hr_1` database, which stores information
    about employees, departments, job roles, locations, and countries, as well as employment history and
    organizational structure."}',
 '{"function_987": "Get the employee_id and salary of employees who report to a manager with a specific
    first name. The data is retrieved from a human resources management system database."}',
 '{"function_1460": "Get the salary and manager_id of employees who do not report to a specific manager.
    The data is retrieved from the `hr_1` database, a human resources management system for a
    multinational corporation."}',
 '{"function_766": "Get the phone numbers of employees with salaries within a specified range. The data
    is sourced from a human resources management system for a multinational corporation."}',
 '{"function_3979": "Get employee information from the `hr_1` database where the salary falls within a
    specified range. The `hr_1` database is a human resources management system for a multinational
    corporation, storing information about employees, departments, job roles, locations, and countries,
    as well as employment history and organizational structure."}',
 '{"function_906": "Get employee information from the `hr_1` database for employees with a specific salary
    . The `hr_1` database is a human resources management system storing information about employees,
    departments, job roles, locations, and countries."}',
 '{"function_986": "Get the employee_id and salary from the `hr_1` database for employees with a
    manager_id matching the provided input. The `hr_1` database is a human resources management system
    storing information about employees, departments, job roles, locations, and countries, as well as
    employment history and organizational structure."}',
```

```
'{"function_849": "Get employee information from the `hr_1` database based on salary range, commission
    percentage, and department ID. The `hr_1` database stores information about employees, departments,
    job roles, locati...
<thought> I can use function_987 to get the employee_id and salary of employees who report to a manager
    with a specific first name. I should find out more about function_987 using the get_info tool. </
    thought>

<execute>
get_info(tool_name="function_987")
</execute>

Observation:
{'type': 'function',
 'function': {'name': 'function_987',
  'description': 'Get the employee_id and salary of employees who report to a manager with a specific
    first name. The data is retrieved from a human resources management system database.',
  'parameters': {'type': 'object',
   'properties': {'lambda_epsilon': {'type': 'str',
    'description': 'The first name of the manager to query'}},
   'required': ['lambda_epsilon']}}}

<thought> I can call function_987 with the lambda_epsilon parameter set to "Shelley" to get the
    employee_id and salary of employees who report to Shelley. </thought>

<execute>
result = function_987(lambda_epsilon="Shelley")
print(result)
</execute>

Observation:
Traceback (most recent call last):

  Cell In[1], line 1
    result = function_987(lambda_epsilon="Shelley")

  File ~/sandbox_utils.py:42831 in function_987
    raise ValueError(f"function_987 is currently unavailable. Please try a different function.")

ValueError: function_987 is currently unavailable. Please try a different function.

<thought> It seems function_987 is currently unavailable. I should try a different function. I can use
    function_986 to get the employee_id and salary of employees with a specific manager_id. I should
    find out more about function_986 using the get_info tool. </thought>

<execute>
get_info(tool_name="function_986")
</execute>

Observation:
{'type': 'function',
 'function': {'name': 'function_986',
  'description': 'Get the employee_id and salary from the `hr_1` database for employees with a manager_id
    matching the provided input. The `hr_1` database is a human resources management system storing
    information about employees, departments, job roles, locations, and countries, as well as employment
     history and organizational structure.',
  'parameters': {'type': 'object',
   'properties': {'beta_epsilon': {'type': 'list',
    'description': 'A list of manager_ids to query'}},
   'required': ['beta_epsilon']}}}

<thought> I can call function_986 with the beta_epsilon parameter set to the manager_id of Shelley.
    However, I don't have the manager_id of Shelley. I need to find a function that can get the
    manager_id of an employee by their first name. I can use the search_tools function to find relevant
    tools. </thought>

<execute>
search_tools(query="employee manager id", num_results=9)
</execute>

Observation:
['{"function_1628": "Get information about employees who do not belong to departments with managers
    within a specified range of manager IDs. The data is retrieved from a human resources management
    system database."}',
 '{"function_987": "Get the employee_id and salary of employees who report to a manager with a specific
    first name. The data is retrieved from a human resources management system database."}',
 '{"function_1626": "Get the department_id from the `departments` table for a range of manager_id values.
    The departments table is part of a human resources management system for a multinational corporation,
     storing information about employees, departments, job roles, locations, and countries, as well as
    employment history and organizational structure."}',
 '{"function_3038": "Get the employee_id and job_title for employees in a specific department. The data
    is sourced from a human resources management system for a multinational corporation."}',
 '{"function_2390": "Get the manager with the most direct reports from the `store_1` database. The store_1
    database manages information about a music store, including employees and their reporting
    relationships."}',
```

```
'{"function_588": "Get the employee_id from the `cre_Doc_Tracking_DB` database for employees with
    specific roles. The database manages and tracks documents, including their types, storage locations,
    and destruction schedules, while also storing information about employees and their roles within an
    organization."}',
'{"function_986": "Get the employee_id and salary from the `hr_1` database for employees with a
    manager_id matching the provided input. The `hr_1` database is a human resources management system
    storing information about employees, departments, job roles, locations, and countries, as well as
    employment history and organizational structure."}',
'{"function_3103": "Get the first name and job ID of employees in a specific department. The data is
    retrieved from a human resources management system database."}',
'{"function_113": "Get...
<thought> I can use function_987 to get the manager_id of Shelley. However, function_987 is currently
    unavailable. I should try a different function. I can use function_1626 to get the department_id for
    a range of manager_id values. However, I don't have the manager_id of Shelley. I need to find a
    function that can get the manager_id of an employee by their first name. I can use the search_tools
    function to find relevant tools. </thought>

<execute>
search_tools(query="employee manager id first name", num_results=9)
</execute>

Observation:
['{"function_987": "Get the employee_id and salary of employees who report to a manager with a specific
    first name. The data is retrieved from a human resources management system database."}',
'{"function_3103": "Get the first name and job ID of employees in a specific department. The data is
    retrieved from a human resources management system database."}',
'{"function_3201": "Get the first name, last name, and hire date of employees in the same department as
    a given employee. The data is sourced from a human resources management system for a multinational
    corporation."}',
'{"function_985": "Get the employee_id from the `employees` table for a specific first_name. The
    employees table stores information about employees in a multinational corporation\'s human resources
    management system"}',
'{"function_303": "Get the department_id of an employee from the `hr_1` database based on their first
    name. The `hr_1` database is a human resources management system storing information about employees,
    departments, job roles, locations, and countries."}',
'{"function_1819": "Get employee information from the `hr_1` database for employees with a matching first
    name. The hr_1 database stores information about employees, departments, job roles, locations, and
    countries, as well as employment history and organizational structure."}',
'{"function_1628": "Get information about employees who do not belong to departments with managers
    within a specified range of manager IDs. The data is retrieved from a human resources management
    system database."}',
'{"function_576": "Get employee information from the `hr_1` database for employees with a matching first
    name. The `hr_1` database is a human resources management system storing information about employees,
    departments, job roles, locations, and countries."}',
'{"function_3038": "Get the employee_id and job_title for employees in a specific department. The data
    is sourced from a human resources management system for a multinational corporation."}']

<thought> I can use function_985 to get the employee_id of Shelley. I should find out more about
    function_985 using the get_info tool. </thought>

<execute>
get_info(tool_name="function_985")
</execute>

Observation:
{'type': 'function',
 'function': {'name': 'function_985',
 'description': "Get the employee_id from the `employees` table for a specific first_name. The employees
    table stores information about employees in a multinational corporation's human resources management
    system",
 'parameters': {'type': 'object',
  'properties': {'lambda_sigma': {'type': 'str',
   'description': 'The first_name to query'}},
  'required': ['lambda_sigma']}}}

<thought> I can call function_985 with the lambda_sigma parameter set to "Shelley" to get the employee_id
    of Shelley. </thought>

<execute>
shelley_id = function_985(lambda_sigma="Shelley")
print(shelley_id)
</execute>

Observation:
[{'EMPLOYEE_ID': 205}]

<thought> I have the employee_id of Shelley, which is 205. I can now call function_986 with the
    beta_epsilon parameter set to [205] to get the employee_id and salary of employees who report to
    Shelley. </thought>

<execute>
result = function_986(beta_epsilon=[205])
print(result)
</execute>
```

```
Observation:
[{'EMPLOYEE_ID': 206, 'SALARY': 8300}]

<solution>
solution = result
</solution>
```

