# OpenReview forum: "Hell or High Water: Evaluating Agentic Recovery from External Failures"
_colmweb.org/COLM/2025/Conference — COLM 2025_

### Official Review · Reviewer_3TTA · 2025-05-10

**Rating:** 5
**Confidence:** 4
**Ethics Flag:** 1

**Summary:**

This paper investigates the ability of language model agents to formulate and execute backup plans when their primary plan fails due to external errors such as unavailable APIs. The authors introduce a benchmark called Hell-or-High-Water, built from a transformed text-to-SQL dataset, where each task has multiple valid solution paths involving API-like function calls. They evaluate open-weight and commercial models and find that current LLM agents struggle significantly with fault-tolerant planning. Even when failure is explicitly communicated, agents often fail to adapt or find alternate solutions.

**Questions To Authors:**

- Why do LLMs fail even when the error is explicit and the backup path is simple? Is it due to instruction-following limitations, planning capacity, or search strategy?
- What are the effects of different prompting techniques? Have you tried ReAct vs. Tree-of-Thought vs. CoT? Any empirical insights? Did you try prompting similar to "if a tool failed, try another tool"?
- How generalizable is this benchmark to real-world tools? For example, can this method be used with actual RESTful APIs rather than synthetic SQL wrappers?
- Memory should be able to be added in CodeAct framework. Does incorporating memory or trajectory replay help with backup planning? Would a more persistent context help agents recognize and avoid repeating failed paths?

**Reasons To Accept:**

- The paper introduces a new and practical benchmark to evaluate backup planning.
-  It addresses a crucial capability gap in LLM agents—the ability to recover from failures, which is vital for robust deployment.

**Reasons To Reject:**

- Limited agent framework been explored. There are many ways to improve the ReAct, reflection, and even too search methods. These are not explored in this paper. The observed failure is based solely on the CodeAct framework.
- The effectiveness of reasoning-heavy models or different prompting styles (e.g., self-reflection, tree-of-thought) is only briefly speculated on rather than rigorously tested.
- Dataset, although clean, does not fully reflect the messiness or ambiguity of real-world APIs.
- Many recent benchmarks already test tool-use and reasoning; this paper’s unique contribution lies solely in the forced failure and multi-path recovery, which could be seen as an incremental extension.

---

> ### Author Response · Authors · 2025-06-01
> **Author Response**
>
> We thank the reviewer for highlighting the practical importance of our work (“a new and practical benchmark to evaluate backup planning” + “addresses a crucial capability gap in LLM agents—the ability to recover from failures, which is vital for robust deployment”).  We address the main concerns below.
>
> > Limited agent framework been explored/What are the effects of different prompting techniques?
>
> We considered a broad range of prompting frameworks in our pilot experiments. We eliminated those approaches that were either not competitive or were effectively equivalent to our setup.
> CodeAct outperforms purely text-based approaches (i.e., ReAct, CoT) in tool use settings (Wang et al., 2024), while being more efficient than expensive searches such as tree of thoughts.
>
> In addition, our implementation of CodeAct bears similarity to Reflexion and can be considered an adaptation of Reflexion as well. As demonstrated in Appendix E, the language model agent naturally generates reflections of its actions when system prompt instructions are considered in conjunction with environmental feedback.
>
> In Section 4.2, we also experiment with different methods of tool retrieval, including long-context and an IR setting similar to APIBank. The results of these experiments are provided in Table 1 and Figure 3.
>
> Please let us know if there is any specific prompting framework in mind that is clearly distinct from our current implementation and likely to perform better than our current ones.
>
> > How generalizable is this benchmark to real-world tools?
>
> Thanks for the question.
>
>
> We note that our aim is *not* to simulate every challenge associated with tool use, but to perform a controlled study of backup planning and agent robustness to external errors.
> Nevertheless, we believe that our benchmark captures several key real-world challenges:
> - unreliable tools and external errors
> - having thousands of tools across many domains and functionalities, which interact with over a hundred different databases from a comprehensive set of domains.
>
> In addition, our documentation follows the format specified by OpenAI function calling which is increasingly becoming industry standard. We use a remote code execution sandbox to run our tools in order to give real compiler feedback and stack traces to the agent.
>
> Tools in our benchmark are qualitatively comparable to real Rest API calls such as the ones found in Toolbench, and vary in terms of complexity, number of arguments, and return types.
>
> > can this method be used with actual RESTful APIs rather than synthetic SQL wrappers?
>
> Thanks for the suggestion. Real world APIs provide no notion of functional correctness or backup plans. The chance that there are redundant/multiple ways of doing the same thing with the available APIs is low and coincidental. We would be unable to check answers on the basis of functional correctness.
>
> > Memory should be able to be added in CodeAct framework.
>
> While we agree that persistent memory and lifelong learning is an important next-step in the improvement of agentic capabilities, the purpose of our benchmark is to enable research in improving backup planning. We therefore leave exploration of persistent memory to future research.
>
> We also highlight that our context window itself serves as a form of episodic memory for the agent, in which past attempts, mistakes, and reflections are kept in-context over the course of solving a problem for the benefit of the agent.
>
> > Why do LLMs fail even when the error is explicit and the backup path is simple? Is it due to instruction-following limitations, planning capacity, or search strategy?
>
>
>
> This is a great question. While fully understanding _why_ LLMs do poorly here may be beyond the scope of the paper, per the suggestion, we will analyze and categorize agent trajectories using the following taxonomy:
> - search failure: correct tool never shows up in results
> - identification failure: correct tool shows up in results, but agent never uses it
> - feedback failure: the agent tries the correct tool, but gives up on using it correctly
> - chaining failure (multi-tool only): the agent identifies one tool but not the other, or uses tools out of order
> - other
>
> Results will be provided as soon as they are available. We hope these additional results will help answer your questions.
>
> __Sources:__
> Executable Code Actions Elicit Better LLM Agents
> _Xingyao Wang, Yangyi Chen, Lifan Yuan, Yizhe Zhang, Yunzhu Li, Hao Peng, Heng Ji_

---

> > ### Comment · Reviewer_3TTA · 2025-06-06
> >
> > Thank you for your response. Some of my concerns have been addressed. Will increase my score.

---

> ### Author Response · Authors · 2025-06-10
> **[Updated] Author Response**
>
> Thank you for your patience. Per some of the suggestions, we have collected results for our taxonomy, compared the difficulty of single- vs multi- function call settings, determined whether different models were getting the same questions wrong, and counted the number of tokens per problem. We have taken our time to ensure that the results are of proper quality.
>
> We provide results for the taxonomy proposed earlier. To reiterate, we define each of the categories as follows:
> - search failure: correct tool never shows up in search results
> - ID failure: correct tool shows up in search results, but agent never uses it
> - chaining failure: the agent identifies one tool but not the other, or uses tools out of order
> - tool use failure: the agent tries the correct tool, but gives up on using it correctly
>
> Overall, we find that across all models most of the failure types were tool ID failures or chaining failures. Tool search and tool-use did not account for a majority of failures. Proprietary models suffered from chaining errors the most, while newer open source models suffered more from tool ID failures. While high rates of tool ID failure may raise questions about the quality of documentation, we address this point by referring to the results of Figure 3, where some models are able to achieve up to 90% accuracy on closed-world versions of the task.
>
> |    Model      |  # of Failures |  Search Failure  | ID Failure  |  Chaining Failure  |  Tool Use Failure |
> | ------------ | -------------- | ------------------ | ----------- | --------------------- | ------------------- |
> |  gemini 2    |       489          |         24.3%         |   27.0%     |      37.0%              |   11.7%               |
> |     gpt 4o     |       511          |         17.4%         |   36.6%     |      39.1%              |   6.85%               |
> |   qwen 72b |       496          |         30.0%         |   43.8%     |      26.2%              |   0.00%               |
> |   llama 3.3  |       495          |         28.5%         |   41.2%     |      30.3%              |   0.00%               |
> |   llama 3.1  |       637          |         38.6%         |   37.5%     |      23.7%              |   0.16%               |
>
> We show that there is largely little performance asymmetry between single vs multiple function calls, suggesting that the number of function calls involved in the available solution was not a confounding variable. For the results in Table 1, we stratify accuracy based on the number of function calls involved in the available solution, and find that, if anything, single call solutions were more challenging for a few models (Llama). Otherwise, we did not observe much performance asymmetry. In addition to the accuracy, we provide the ratio of single function calls to multi function calls and show that ratio is fairly balanced.
>
> |    Model      | Single Call | Multi Call |      Ratio (Single:Multi)     |
> | ------------ | ------------ | --------- | ------------ |
> |  gemini 2    |        .383     |     .432     |   355:475    |
> |    gpt-4o      |       .399      |     .373    |    358:472   |
> |   qwen 72B |       .389      |     .418    |    445:385   |
> |   llama-3.3  |       .300      |     .495    |    390:400   |
> |   llama-3.1  |       .145      |     .400    |    545:285   |
>
> We wanted to see if different models were getting the same questions wrong in Table 1. We calculated Fleiss’s Kappa to see whether the different models “agreed” on the incorrect questions. Our results indicated an agreement of .246, showing “fair” but somewhat low agreement and suggesting that different models tend to get different questions wrong. In other words, model behavior rather than inherent data uncertainty accounts for our results.
>
> We provide statistics on the average token budget per problem. As a comparison, we turn to the human demonstrations in the Webshop benchmark, which averaged to 2746 tokens. Since these are roughly in the same order of magnitude as our benchmark, we believe that our benchmark is comparable to Webshop in terms of compute cost. Additionally, the Webshop token average did not factor-in the observation of HTML, which would lead to a higher value in practice.
>
> |    Model      | External Error (N) | External Error (Y) |
> | ------------ | -------------------- | --------------------- |
> | gpt 4o         |         6102            |            7360           |
> | qwen 72b   |         6495             |            8410           |
> | llama 3.3    |         6802            |            8468           |
> | llama 3.1    |         7337            |            8437           |

---

> ### Author Response · Authors · 2025-06-10
> **[Updatedx2] Author Response**
>
> In the closed-world setting, to better ascertain the causes of the performance decrease when search space is increased, we apply our taxonomy to the Gemini-2 results. We find that as the search space is increased, the share of identification failures increases from 32.7% at 3 tools to 54.8% at 81 tools, suggesting that failing to recognize the correct tool(s) when provided is the leading cause of this performance decrease. These results also suggest that tool chaining, or the ability to compose the right function calls in order, is perhaps secondary to the challenge of identifying them in the first place.
>
> |   # of tools in context  |  # of Failures  | ID Failure  |  Chaining Failure  |  Tool Use Failure |
> | ------------------------- | -------------- | ----------- | --------------------- | ------------------- |
> |             3                     |       211          |   32.7%     |      16.1%              |   51.2%               |
> |             9                     |       332          |   45.8%     |      23.8%              |   30.4%               |
> |             27                   |       451          |   54.8%     |      16.9%              |   28.4%               |
> |             81                   |       510          |   54.9%     |      15.5%              |   29.6%               |
>
> In the process of conducting this analysis, we refined the taxonomic analysis (bug fixes and other improvements), leading to updated results. We provide the updated table below:
>
> |    Model      |  # of Failures |  Search Failure  | ID Failure  |  Chaining Failure  |  Tool Use Failure |
> | ------------ | -------------- | ------------------ | ----------- | --------------------- | ------------------- |
> |  gemini 2    |       489          |         57.1%         |   25.5%     |      12.3%              |   6.14%               |
> |     gpt 4o     |       511          |         52.6%         |   24.6%     |      8.61%              |   14.1%               |
> |   qwen 72b |       496          |         66.3%         |   11.9%     |      13.9%              |   7.86%               |
> |   llama 3.3  |       495          |         59.3%         |   18.0%     |      11.7%              |   10.9%               |
> |   llama 3.1  |       637          |         60.4%         |   26.4%     |      7.06%              |   6.12%               |
>
>
> Search failures account for between 53% and 60.4% of the failures, followed by tool identification failures.
>
> Note that the agents have the ability to compose and refine search queries. Search errors may be explained in part by a failure to adequately compose search queries reflecting the required information -- a critical capability for agents interacting with large search spaces. Our benchmark is designed to evaluate this capability and comprises a large search space of 4450 functions. Therefore, our findings highlight an important gap in the capabilities of current-generation LLMs in interacting with search tools to identify solutions to non-trivial problems.

---

### Official Review · Reviewer_DGtT · 2025-05-11

**Rating:** 7
**Confidence:** 4
**Ethics Flag:** 1

**Summary:**

Hell or High Water introduces the first benchmark explicitly designed to test whether language-model agents can recover from failures when their initial tool-use plans break. The authors constructed 830 text-to-SQL tasks from the Spider dataset, each solvable via two distinct tool paths, and systematically disabled agents' initial tool choices at runtime to test adaptive replanning. Evaluations across leading LLMs demonstrated significant drops in performance when initial plans failed, highlighting critical robustness gaps. This benchmark uniquely stresses forced plan adaptation, exposing limitations in current state-of-the-art agent models.

**Questions To Authors:**

- Does the benchmark control for tool naming conventions (e.g., meaningful vs. obfuscated names)? Could tool discovery be unfairly easy if names leak intent?
- The name Hell or High Water is memorable, but it might benefit from a more descriptive subtitle for searchability.

**Reasons To Accept:**

- The paper addresses a critical yet underexplored capability in tool-using agents—robust recovery from failure, which could be a very interesting topic to research community.
- The paper is well-written and carefully structured, with clear definitions, illustrative figures, and comprehensive experimental analysis.
- The benchmark is well-engineered, with 4,450 auto-generated, type-checked tools, a clean interface, and full automation of grading.

**Reasons To Reject:**

- The paper reports aggregate performance drops when the primary tool fails, but offers limited qualitative or taxonomic analysis of how or why agents fail to recover (e.g., search failures, chaining errors, hallucination, etc.).
- While the authors use semantic search over 4,000 tools, it remains unclear how feasible or costly it is to run this benchmark at scale in terms of latency, inference tokens, or search efficiency.

---

> ### Author Response · Authors · 2025-06-02
> **Author Response**
>
> Thanks for highlighting the clarity of our writing, the quality of our benchmark's engineering and interface, and its exploration of a "critical yet underexplored capability in tool-using agents." We address the main concerns below.
>
> > limited qualitative or taxonomic analysis
>
> This is a great question. Per the suggestion, we will analyze and categorize agent trajectories using the following taxonomy:
> - search failure: correct tool never shows up in results
> - identification failure: correct tool shows up in results, but agent never uses it
> - feedback failure: the agent tries the correct tool, but gives up on using it correctly
> - chaining failure (multi-tool only): the agent identifies one tool but not the other, or uses tools out of order
> - other
>
> Results will be provided as soon as they are available. Please let us know if there are other categories worth including
>
> > unclear how feasible or costly it is to run this benchmark at scale
>
> Thanks for the suggestion. Despite the number of tools, searching for tools is relatively efficient since it is information retrieval on a vector DB. Regarding latency and number of tokens, our implementation uses VLLM, an efficient state of the art inference engine that is industry standard and well benchmarked. We are currently counting the number of tokens used per problem to provide a notion of computation cost and will report these numbers once available.
>
> > Does the benchmark control for tool naming conventions ... Could tool discovery be unfairly easy if names leak intent?
>
> In the real world, functions may or may not have meaningful names. To control for this variability and remove the potential confounding factor of function naming, we make all function names obfuscated. However, we make up for this by providing meaningful documentation to aid the agent in determining the functionality of a function.

---

> > ### Comment · Reviewer_DGtT · 2025-06-04
> > **Thanks for the response.**
> >
> > I appreciate the authors’ plans to introduce a taxonomy of failure modes and conduct trajectory-level analysis. This addition will significantly enhance the diagnostic value of the benchmark and help the community better understand where and why current agents fall short.
> >
> > I would like to raise my score once the failure taxonomy analysis and token cost results are reported.
> >
> > Looking forward to the updated results.

---

> ### Author Response · Authors · 2025-06-10
> **[Updated] Author Response**
>
> Thank you for your patience. Per some of the suggestions, we have collected results for our taxonomy, compared the difficulty of single- vs multi- function call settings, determined whether different models were getting the same questions wrong, and counted the number of tokens per problem. We have taken our time to ensure that the results are of proper quality.
>
> We provide results for the taxonomy proposed earlier. To reiterate, we define each of the categories as follows:
> - search failure: correct tool never shows up in search results
> - ID failure: correct tool shows up in search results, but agent never uses it
> - chaining failure: the agent identifies one tool but not the other, or uses tools out of order
> - tool use failure: the agent tries the correct tool, but gives up on using it correctly
>
> Overall, we find that across all models most of the failure types were tool ID failures or chaining failures. Tool search and tool-use did not account for a majority of failures. Proprietary models suffered from chaining errors the most, while newer open source models suffered more from tool ID failures. While high rates of tool ID failure may raise questions about the quality of documentation, we address this point by referring to the results of Figure 3, where some models are able to achieve up to 90% accuracy on closed-world versions of the task.
>
> |    Model      |  # of Failures |  Search Failure  | ID Failure  |  Chaining Failure  |  Tool Use Failure |
> | ------------ | -------------- | ------------------ | ----------- | --------------------- | ------------------- |
> |  gemini 2    |       489          |         24.3%         |   27.0%     |      37.0%              |   11.7%               |
> |     gpt 4o     |       511          |         17.4%         |   36.6%     |      39.1%              |   6.85%               |
> |   qwen 72b |       496          |         30.0%         |   43.8%     |      26.2%              |   0.00%               |
> |   llama 3.3  |       495          |         28.5%         |   41.2%     |      30.3%              |   0.00%               |
> |   llama 3.1  |       637          |         38.6%         |   37.5%     |      23.7%              |   0.16%               |
>
> We show that there is largely little performance asymmetry between single vs multiple function calls, suggesting that the number of function calls involved in the available solution was not a confounding variable. For the results in Table 1, we stratify accuracy based on the number of function calls involved in the available solution, and find that, if anything, single call solutions were more challenging for a few models (Llama). Otherwise, we did not observe much performance asymmetry. In addition to the accuracy, we provide the ratio of single function calls to multi function calls and show that ratio is fairly balanced.
>
> |    Model      | Single Call | Multi Call |      Ratio (Single:Multi)     |
> | ------------ | ------------ | --------- | ------------ |
> |  gemini 2    |        .383     |     .432     |   355:475    |
> |    gpt-4o      |       .399      |     .373    |    358:472   |
> |   qwen 72B |       .389      |     .418    |    445:385   |
> |   llama-3.3  |       .300      |     .495    |    390:400   |
> |   llama-3.1  |       .145      |     .400    |    545:285   |
>
> We wanted to see if different models were getting the same questions wrong in Table 1. We calculated Fleiss’s Kappa to see whether the different models “agreed” on the incorrect questions. Our results indicated an agreement of .246, showing “fair” but somewhat low agreement and suggesting that different models tend to get different questions wrong. In other words, model behavior rather than inherent data uncertainty accounts for our results.
>
> We provide statistics on the average token budget per problem. As a comparison, we turn to the human demonstrations in the Webshop benchmark, which averaged to 2746 tokens. Since these are roughly in the same order of magnitude as our benchmark, we believe that our benchmark is comparable to Webshop in terms of compute cost. Additionally, the Webshop token average did not factor-in the observation of HTML, which would lead to a higher value in practice.
>
> |    Model      | External Error (N) | External Error (Y) |
> | ------------ | -------------------- | --------------------- |
> | gpt 4o         |         6102            |            7360           |
> | qwen 72b   |         6495             |            8410           |
> | llama 3.3    |         6802            |            8468           |
> | llama 3.1    |         7337            |            8437           |

---

> > ### Comment · Reviewer_DGtT · 2025-06-10
> > **Thanks for the updated results.**
> >
> > Thanks the authors for the updated results. Correspondingly, I have raised my score.

---

> ### Author Response · Authors · 2025-06-10
> **[Updatedx2] Author Response**
>
> In the closed-world setting, to better ascertain the causes of the performance decrease when search space is increased, we apply our taxonomy to the Gemini-2 results. We find that as the search space is increased, the share of identification failures increases from 32.7% at 3 tools to 54.8% at 81 tools, suggesting that failing to recognize the correct tool(s) when provided is the leading cause of this performance decrease. These results also suggest that tool chaining, or the ability to compose the right function calls in order, is perhaps secondary to the challenge of identifying them in the first place.
>
> |   # of tools in context  |  # of Failures  | ID Failure  |  Chaining Failure  |  Tool Use Failure |
> | ------------------------- | -------------- | ----------- | --------------------- | ------------------- |
> |             3                     |       211          |   32.7%     |      16.1%              |   51.2%               |
> |             9                     |       332          |   45.8%     |      23.8%              |   30.4%               |
> |             27                   |       451          |   54.8%     |      16.9%              |   28.4%               |
> |             81                   |       510          |   54.9%     |      15.5%              |   29.6%               |
>
> In the process of conducting this analysis, we refined the taxonomic analysis (bug fixes and other improvements), leading to updated results. We provide the updated table below:
>
> |    Model      |  # of Failures |  Search Failure  | ID Failure  |  Chaining Failure  |  Tool Use Failure |
> | ------------ | -------------- | ------------------ | ----------- | --------------------- | ------------------- |
> |  gemini 2    |       489          |         57.1%         |   25.5%     |      12.3%              |   6.14%               |
> |     gpt 4o     |       511          |         52.6%         |   24.6%     |      8.61%              |   14.1%               |
> |   qwen 72b |       496          |         66.3%         |   11.9%     |      13.9%              |   7.86%               |
> |   llama 3.3  |       495          |         59.3%         |   18.0%     |      11.7%              |   10.9%               |
> |   llama 3.1  |       637          |         60.4%         |   26.4%     |      7.06%              |   6.12%               |
>
>
> Search failures account for between 53% and 60.4% of the failures, followed by tool identification failures.
>
> Note that the agents have the ability to compose and refine search queries. Search errors may be explained in part by a failure to adequately compose search queries reflecting the required information -- a critical capability for agents interacting with large search spaces. Our benchmark is designed to evaluate this capability and comprises a large search space of 4450 functions. Therefore, our findings highlight an important gap in the capabilities of current-generation LLMs in interacting with search tools to identify solutions to non-trivial problems.

---

### Official Review · Reviewer_swx9 · 2025-05-12

**Rating:** 6
**Confidence:** 4
**Ethics Flag:** 1

**Summary:**

This paper explores a relatively under-studied but practically important capability of LLM agents: their ability to formulate and execute backup plans when their initial strategy fails due to external errors. To this end, the authors convert a text-to-SQL dataset (Spider) into a planning environment with callable functions and design an interactive benchmark ("Hell-or-High-Water") that systematically disables the agent's initial plan by disabling the key function, prompting the model to recover with an alternative. The benchmark includes 830 questions and 4450 functions, each with at least two solution paths. The authors find that across both open-weight and commercial LLMs, model performance drops significantly under such error conditions , indicating a clear limitation in current models' ability to recover from failure. They also investigate the impact of model scale and question underspecification on performance. The study provides useful insights into improving the robustness of LLM agents.

**Questions To Authors:**

1. Since the backup plans often involve multiple function calls while the initial plan typically uses only one, could this asymmetry affect the observed accuracy drop and confound the comparison between primary and backup strategies?
2. Have you considered including alternative plans that are logically distinct from the original, rather than just decompositions? Would such designs offer a more comprehensive test of backup planning ability?
3. Your closed-world experiments (Figure 3) indicate that when backup plans are necessary, performance drops as distractor functions increase. This suggests a key difficulty lies in finding alternative functions during replanning. Could you elaborate on why this search capability seems particularly impaired after an initial plan failure？
4. You briefly mention that preliminary experiments with thinking models suggested they might require different prompting strategies and that cost precluded a more detailed analysis. Could you provide a little more insight into what initial differences you observed, even if anecdotal?

**Reasons To Accept:**

- The "Hell-or-High-Water" benchmark, which focuses on the capability of LLM agents to formulate backup plans, is reasonably and novelly designed with multiple solution paths and a large function search space. Although some limitations still exist, its advanced benchmark construction scheme nonetheless provides adequate support for exploring this scenario and facilitates subsequent research.
- The paper reveals that current LLMs, even when faced with explicit error messages, exhibit significant difficulty in formulating and executing backup plans. This offers the latest insights into how to enhance the robustness of LLM agents.
- The experimental section also achieves in-depth exploration, not only covering a variety of LLMs but also further investigating the impact of factors such as search space size, model scale, and question underspecification on the results.

**Reasons To Reject:**

The experimental setup and backup plan design are limited in scope: backup solutions are typically composed of multiple sub-functions derived from decomposing the initial single-function plan, which may introduce a structural bias and uneven difficulty between primary and backup paths. This restricts the generality of the benchmark and fails to fully capture more diverse or logically distinct backup strategies.

---

> ### Author Response · Authors · 2025-06-02
> **Author Response Continued**
>
> > Your closed-world experiments (Figure 3) indicate that when backup plans are necessary, performance drops as distractor functions increase. This suggests a key difficulty lies in finding alternative functions during replanning. Could you elaborate on why this search capability seems particularly impaired after an initial plan failure？
>
> Thanks for the question. Increasing the number of available tools in-context effectively increases the size of the search space. If language model agents express high confidence for only the original plan, then this combined with the size of the search space would contribute to the impaired search capability after an initial plan failure.
>
> We provide a brief example illustrating the effect of increased search space:
>
> When given the problem _What is the total revenue of companies with revenue greater than the lowest revenue of any manufacturer in Paris?_ (correct tools: 756 OR 659 + 755)---
>
> The agent (qwen) correctly identifies the backup plan in the simplest setting of 3 available tools:
>
> > [AGENT] To solve this, I need to first find the minimum revenue of any manufacturer in Paris using function_659. Then I will use that value as input to function_755 to get the total revenue of companies with revenue greater than that minimum.
>
> …
>
> > [AGENT] function_659 is unavailable. I will try using function_756 instead, which takes the headquarter as input and returns the total revenue of manufacturers with revenue greater than the minimum revenue of manufacturers in that headquarter.
>
> However, for the same problem on the slightly more difficult setting of 9 available tools, the agent fails to identify the correct sequence of tools to call.
>
> > [AGENT] To solve this, I need to first find the lowest revenue of any manufacturer in Paris using function_659. Then, I will use that value to find the total revenue of companies with revenue greater than this lowest revenue using function_755.
>
> > [AGENT] It seems that function_659 is currently unavailable. I will use function_3318 to get the total revenue of manufacturers in Paris, and then use function_755 to get the total revenue of companies with revenue greater than the minimum revenue. However, I need to find a way to get the minimum revenue from the total revenue of manufacturers in Paris. I will first get the total revenue of manufacturers in Paris.
>
> …
>
> > [AGENT] The total revenue of companies with revenue greater than the average revenue (which is used as a proxy for the minimum revenue) is 450.0. However, this is not the exact solution, as the average revenue is not the same as the minimum revenue. I will note this in the solution.
>
> We hope that our taxonomical analysis will further explain these results as well.
>
> > You briefly mention that preliminary experiments with thinking models suggested they might require different prompting strategies and that cost precluded a more detailed analysis. Could you provide a little more insight into what initial differences you observed, even if anecdotal?
>
> Thinking models responded poorly to environmental feedback and did not consistently follow guidelines for producing structured output, potentially hinting at worse instruction following abilities than non-thinking models (Li et al., 2025, Fu et al., 2025).
>
> For instance, when the following feedback was given
> > [ENVIRONMENT] Observation: AssertionError('no \<solution\> or \<execute\> tags detected! please follow the format specified in the prompt')
>
> Non-thinking models were consistently able to correct their response to the proper format, but thinking models ignored the feedback altogether, responding with:
>
> > [THINKING AGENT] Okay, let's tackle this problem step by step. The user wants the names and salaries of instructors in the Cybernetics department who earn less than the average salary in that department. First, I need to find the average salary of instructors in the Cybernetics department…
>
> ### Sources:
> When Thinking Fails: The Pitfalls of Reasoning for Instruction-Following in LLMs.
> _Xiaomin Li, Zhou Yu, Zhiwei Zhang, Xupeng Chen, Ziji Zhang, Yingying Zhuang, Narayanan Sadagopan, Anurag Beniwal_
>
> Scaling Reasoning, Losing Control: Evaluating Instruction Following in Large Reasoning Models
> _Tingchen Fu, Jiawei Gu, Yafu Li, Xiaoye Qu, Yu Cheng_

---

> > ### Comment · Reviewer_swx9 · 2025-06-04
> > **Thanks for your reply**
> >
> > Thanks to the authors for the clear and thoughtful response. While they did address most points and added some helpful examples, I still feel a few core issues remain:
> >
> > 1. The asymmetry between simple initial plans and multi-step backups may introduce structural bias, and it's unclear whether this was properly controlled for.
> >
> > 2. The lack of logically distinct alternative plans limits the generality of the benchmark — decompositional alternatives aren’t enough to fully capture real-world replanning diversity.
> >
> > 3. The insight into search degradation under distractor functions is interesting, but could benefit from deeper analysis (e.g., are models just overconfident, or fundamentally bad at fallback search?).
> >
> > Overall, I still think the benchmark is valuable and novel, but I maintain my score until some of these design concerns are better addressed.

---

> ### Author Response · Authors · 2025-06-02
> **Author Response**
>
> Thanks for highlighting the novelty of our backup planning benchmark design and that our "advanced benchmark construction scheme ... provides adequate support for exploring this scenario and facilitates subsequent research." We address the main concerns below.
>
> > Since the backup plans often involve multiple function calls while the initial plan typically uses only one, could this asymmetry affect the observed accuracy drop and confound the comparison between primary and backup strategies?
>
> Thanks for the suggestion. To measure the impact of a potential asymmetry, we are running additional experiments to measure performance without external errors. We study two settings: (1) only the tools in the "initial plan" are available, and (2) only the tools in the "backup plan" are available. This way, we eliminate the potential confounding factor of having a distinct number of tools needed to solve the problem. We will provide these results as soon as they are available.
>
> > Have you considered including alternative plans that are logically distinct from the original, rather than just decompositions? Would such designs offer a more comprehensive test of backup planning ability?
>
> Thanks for the suggestion. While we acknowledge that equivalent but logically distinct plans are ideal, we would like to point out that achieving these conditions at scale while ensuring functional correctness remains difficult in practice. The purpose of our benchmark is to show that language model agents have a more challenging time formulating backup plans when faced with external errors. Providing a comprehensive understanding of this phenomenon is orthogonal to our research questions and we leave this work to future research.

---

> ### Author Response · Authors · 2025-06-10
> **[Updated] Author Response**
>
> Thank you for your patience. Per some of the suggestions, we have collected results for our taxonomy, compared the difficulty of single- vs multi- function call settings, determined whether different models were getting the same questions wrong, and counted the number of tokens per problem. We have taken our time to ensure that the results are of proper quality.
>
> We provide results for the taxonomy proposed earlier. To reiterate, we define each of the categories as follows:
> - search failure: correct tool never shows up in search results
> - ID failure: correct tool shows up in search results, but agent never uses it
> - chaining failure: the agent identifies one tool but not the other, or uses tools out of order
> - tool use failure: the agent tries the correct tool, but gives up on using it correctly
>
> Overall, we find that across all models most of the failure types were tool ID failures or chaining failures. Tool search and tool-use did not account for a majority of failures. Proprietary models suffered from chaining errors the most, while newer open source models suffered more from tool ID failures. While high rates of tool ID failure may raise questions about the quality of documentation, we address this point by referring to the results of Figure 3, where some models are able to achieve up to 90% accuracy on closed-world versions of the task.
>
> |    Model      |  # of Failures |  Search Failure  | ID Failure  |  Chaining Failure  |  Tool Use Failure |
> | ------------ | -------------- | ------------------ | ----------- | --------------------- | ------------------- |
> |  gemini 2    |       489          |         24.3%         |   27.0%     |      37.0%              |   11.7%               |
> |     gpt 4o     |       511          |         17.4%         |   36.6%     |      39.1%              |   6.85%               |
> |   qwen 72b |       496          |         30.0%         |   43.8%     |      26.2%              |   0.00%               |
> |   llama 3.3  |       495          |         28.5%         |   41.2%     |      30.3%              |   0.00%               |
> |   llama 3.1  |       637          |         38.6%         |   37.5%     |      23.7%              |   0.16%               |
>
> We show that there is largely little performance asymmetry between single vs multiple function calls, suggesting that the number of function calls involved in the available solution was not a confounding variable. For the results in Table 1, we stratify accuracy based on the number of function calls involved in the available solution, and find that, if anything, single call solutions were more challenging for a few models (Llama). Otherwise, we did not observe much performance asymmetry. In addition to the accuracy, we provide the ratio of single function calls to multi function calls and show that ratio is fairly balanced.
>
> |    Model      | Single Call | Multi Call |      Ratio (Single:Multi)     |
> | ------------ | ------------ | --------- | ------------ |
> |  gemini 2    |        .383     |     .432     |   355:475    |
> |    gpt-4o      |       .399      |     .373    |    358:472   |
> |   qwen 72B |       .389      |     .418    |    445:385   |
> |   llama-3.3  |       .300      |     .495    |    390:400   |
> |   llama-3.1  |       .145      |     .400    |    545:285   |
>
> We wanted to see if different models were getting the same questions wrong in Table 1. We calculated Fleiss’s Kappa to see whether the different models “agreed” on the incorrect questions. Our results indicated an agreement of .246, showing “fair” but somewhat low agreement and suggesting that different models tend to get different questions wrong. In other words, model behavior rather than inherent data uncertainty accounts for our results.
>
> We provide statistics on the average token budget per problem. As a comparison, we turn to the human demonstrations in the Webshop benchmark, which averaged to 2746 tokens. Since these are roughly in the same order of magnitude as our benchmark, we believe that our benchmark is comparable to Webshop in terms of compute cost. Additionally, the Webshop token average did not factor-in the observation of HTML, which would lead to a higher value in practice.
>
> |    Model      | External Error (N) | External Error (Y) |
> | ------------ | -------------------- | --------------------- |
> | gpt 4o         |         6102            |            7360           |
> | qwen 72b   |         6495             |            8410           |
> | llama 3.3    |         6802            |            8468           |
> | llama 3.1    |         7337            |            8437           |

---

> ### Author Response · Authors · 2025-06-10
> **[Updatedx2] Author Response**
>
> In the closed-world setting, to better ascertain the causes of the performance decrease when search space is increased, we apply our taxonomy to the Gemini-2 results. We find that as the search space is increased, the share of identification failures increases from 32.7% at 3 tools to 54.8% at 81 tools, suggesting that failing to recognize the correct tool(s) when provided is the leading cause of this performance decrease. These results also suggest that tool chaining, or the ability to compose the right function calls in order, is perhaps secondary to the challenge of identifying them in the first place.
>
> |   # of tools in context  |  # of Failures  | ID Failure  |  Chaining Failure  |  Tool Use Failure |
> | ------------------------- | -------------- | ----------- | --------------------- | ------------------- |
> |             3                     |       211          |   32.7%     |      16.1%              |   51.2%               |
> |             9                     |       332          |   45.8%     |      23.8%              |   30.4%               |
> |             27                   |       451          |   54.8%     |      16.9%              |   28.4%               |
> |             81                   |       510          |   54.9%     |      15.5%              |   29.6%               |
>
> In the process of conducting this analysis, we refined the taxonomic analysis (bug fixes and other improvements), leading to updated results. We provide the updated table below:
>
> |    Model      |  # of Failures |  Search Failure  | ID Failure  |  Chaining Failure  |  Tool Use Failure |
> | ------------ | -------------- | ------------------ | ----------- | --------------------- | ------------------- |
> |  gemini 2    |       489          |         57.1%         |   25.5%     |      12.3%              |   6.14%               |
> |     gpt 4o     |       511          |         52.6%         |   24.6%     |      8.61%              |   14.1%               |
> |   qwen 72b |       496          |         66.3%         |   11.9%     |      13.9%              |   7.86%               |
> |   llama 3.3  |       495          |         59.3%         |   18.0%     |      11.7%              |   10.9%               |
> |   llama 3.1  |       637          |         60.4%         |   26.4%     |      7.06%              |   6.12%               |
>
>
> Search failures account for between 53% and 60.4% of the failures, followed by tool identification failures.
>
> Note that the agents have the ability to compose and refine search queries. Search errors may be explained in part by a failure to adequately compose search queries reflecting the required information -- a critical capability for agents interacting with large search spaces. Our benchmark is designed to evaluate this capability and comprises a large search space of 4450 functions. Therefore, our findings highlight an important gap in the capabilities of current-generation LLMs in interacting with search tools to identify solutions to non-trivial problems.

---

### Decision · Program_Chairs · 2025-07-08

**Decision:**

Accept

**Comment:**

This paper constructs a dataset to evaluate the ability of LLM-based agents in formulating _backup plans_, i.e., exploring alternative plans when the first plan has failed. The dataset is derived from Spider, a text-to-SQL dataset. Specifically, complete SQL queries from Spider are decomposed and transformed into more atomic Python functions in a deterministic way. Documentation of the functions are generated using an LLM. By construct, each of the tasks has at least two alternative sequences of function calls to complete the goal, originating from different ways of decomposing the original SQL query; this creates the basis for backup planning evaluation. Evaluation on several open and proprietary LLMs show that they suffer a substantial drop when they need to explore backup plans.

Overall, this is an interesting work. Evaluating the ability of agents in exploring backup plans is an important problem, and it's an interesting idea to derive such evaluation tasks from text-to-SQL that provides some desired properties such as reliability in automated evaluation and a large space of functions. On the other hand, there are certainly various structural biases introduced in the dataset construction. A few examples:
1. These SQL-derived Python functions are quite different from typical APIs agents deal with in realistic settings.
2. Even though there are a large number of functions, there could be large semantic overlaps among functions (introduced in an artificial way by construct).
3. The documentation, which in this setting is probably the most important information for LLMs to select from the functions, is automatically generated by an LLM with some limited unit tests; a human evaluation is likely needed.
4. The alternative plans, by construct, have a very specific relationship between each other.

As a first attempt at this type of evaluation, the dataset is still a valuable contribution despite the limitations. But it's important to note these limitations; the current Limitations discussion needs to be expanded.